# REINFORCEMENT LEARNING ALGORITHM SELECTION

**Romain Laroche[1] and Raphaël Féraud[2]**

[1] Microsoft Research, Montréal, Canada

[2] Orange Labs, Lannion, France

## ABSTRACT

This paper formalises the problem of online algorithm selection in the context of Reinforcement Learning (RL). The setup is as follows: given an episodic task and a finite number of off-policy RL algorithms, a meta-algorithm has to decide which RL algorithm is in control during the next episode so as to maximize the expected return. The article presents a novel meta-algorithm, called Epochal Stochastic Bandit Algorithm Selection (ESBAS). Its principle is to freeze the policy updates at each epoch, and to leave a rebooted stochastic bandit in charge of the algorithm selection. Under some assumptions, a thorough theoretical analysis demonstrates its near-optimality considering the structural sampling budget limitations. ESBAS is first empirically evaluated on a dialogue task where it is shown to outperform each individual algorithm in most configurations. ESBAS is then adapted to a true online setting where algorithms update their policies after each transition, which we call SSBAS. SSBAS is evaluated on a fruit collection task where it is shown to adapt the stepsize parameter more efficiently than the classical hyperbolic decay, and on an Atari game, where it improves the performance by a wide margin.

## 1 INTRODUCTION

Reinforcement Learning (RL, Sutton & Barto (1998)) is a machine learning framework for optimising the behaviour of an agent interacting with an unknown environment. For the most practical problems, such as dialogue or robotics, trajectory collection is costly and sample efficiency is the main key performance indicator. Consequently, when applying RL to a new problem, one must carefully choose in advance a model, a representation, an optimisation technique and their parameters. Facing the complexity of choice, RL and domain expertise is not sufficient. Confronted to the cost of data, the popular *trial and error* approach shows its limits.

We develop an *online* learning version (Gagliolo & Schmidhuber, 2006; 2010) of Algorithm Selection (AS, Rice (1976); Smith-Miles (2009); Kotthoff (2012)). It consists in testing several algorithms on the task and in selecting the best one at a given time. For clarity, throughout the whole article, the algorithm selector is called a *meta-algorithm*, and the set of algorithms available to the meta-algorithm is called a *portfolio*. The meta-algorithm maximises an objective function such as the RL return. Beyond the sample efficiency objective, the online AS approach besides addresses four practical problems for online RL-based systems. First, it improves robustness: if an algorithm fails to terminate, or outputs to an aberrant policy, it will be dismissed and others will be selected instead. Second, convergence guarantees and empirical efficiency may be united by covering the empirically efficient algorithms with slower algorithms that have convergence guarantees. Third, it enables curriculum learning: shallow models control the policy in the early stages, while deep models discover the best solution in late stages. And four, it allows to define an objective function that is not an RL return.

A fair algorithm selection implies a fair budget allocation between the algorithms, so that they can be equitably evaluated and compared. In order to comply with this requirement, the reinforcement algorithms in the portfolio are assumed to be *off-policy*, and are trained on every trajectory, regardless which algorithm controls it. Section 2 provides a unifying view of RL algorithms, that allows information sharing between algorithms, whatever their state representations and optimisation techniques. It also formalises the problem of online selection of off-policy RL algorithms.

Next, Section 3 presents the Epochal Stochastic Bandit AS (ESBAS), a novel meta-algorithm addressing the online off-policy RL AS problem. Its principle relies on a doubling trick: it divides the time-scale into epochs of exponential length inside which the algorithms are not allowed to update their policies. During each epoch, the algorithms have therefore a constant policy and a stochastic multi-armed bandit can be in charge of the AS with strong pseudo-regret theoretical guaranties. A thorough theoretical analysis provides for ESBAS upper bounds. Then, Section 4 evaluates ESBAS on a dialogue task where it outperforms each individual algorithm in most configurations.

Afterwards, in Section 5, ESBAS, which is initially designed for a growing batch RL setting, is adapted to a true online setting where algorithms update their policies after each transition, which we call SSBAS. It is evaluated on a fruit collection task where it is shown to adapt the stepsize parameter more efficiently than the classical hyperbolic decay, and on Q*bert, where running several DQN with different network size and depth in parallel allows to improve the final performance by a wide margin. Finally, Section 6 concludes the paper with prospective ideas of improvement.

## 2 Algorithm Selection for RL

### 2.1 Unifying view of RL algorithms

The goal of this section is to enable information sharing between algorithms, even though they are considered as black boxes. We propose to share their trajectories expressed in a universal format: the *interaction process*.

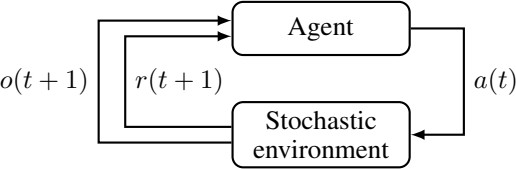

Reinforcement Learning (RL) consists in learning through trial and error to control an agent behaviour in a stochastic environment: at each time step $t \in \mathbb{N}$, the agent performs an action

Figure 1: RL framework: after performing action $a(t)$, the agent perceives observation $o(t+1)$ and receives reward $r(t+1)$.

$a(t) \in \mathcal{A}$, and then perceives from its environment a signal $o(t) \in \Omega$ called observation, and receives a reward $r(t) \in \mathbb{R}$, bounded between $R_{min}$ and $R_{max}$. Figure 1 illustrates the RL framework. This interaction process is not Markovian: the agent may have an internal memory.

In this article, the RL problem is assumed to be episodic. Let us introduce two time scales with different notations. First, let us define *meta-time* as the time scale for AS: at one meta-time $\tau$ corresponds a meta-algorithm decision, *i.e.* the choice of an algorithm and the generation of a full episode controlled with the policy determined by the chosen algorithm. Its realisation is called a *trajectory*. Second, *RL-time* is defined as the time scale inside a trajectory, at one RL-time $t$ corresponds one triplet composed of an observation, an action, and a reward.

Let $\mathscr{E}$ denote the space of trajectories. A *trajectory* $\varepsilon_\tau \in \mathscr{E}$ collected at meta-time $\tau$ is formalised as a sequence of (observation, action, reward) triplets: $\varepsilon_\tau = \langle o_\tau(t), a_\tau(t), r_\tau(t) \rangle_{t \in [\![1, |\varepsilon_\tau|]\!]} \in \mathscr{E}$, where $|\varepsilon_\tau|$ is the length of trajectory $\varepsilon_\tau$. The objective is, given a discount factor $0 \leq \gamma < 1$, to generate trajectories with high discounted cumulative reward, also called *return*, and noted $\mu(\varepsilon_\tau) = \sum_{t=1}^{|\varepsilon_\tau|} \gamma^{t-1} r_\tau(t)$. Since $\gamma < 1$ and $R$ is bounded, the return is also bounded. The *trajectory set* at meta-time $T$ is denoted by $\mathcal{D}_T = \{\varepsilon_\tau\}_{\tau \in [\![1,T]\!]} \in \mathscr{E}^T$. A sub-trajectory of $\varepsilon_\tau$ until RL-time $t$ is called the *history* at RL-time $t$ and written $\varepsilon_\tau(t)$ with $t \leq |\varepsilon_\tau|$. The history records what happened in episode $\varepsilon_\tau$ until RL-time $t$: $\varepsilon_\tau(t) = \langle o_\tau(t'), a_\tau(t'), r_\tau(t') \rangle_{t' \in [\![1,t]\!]} \in \mathscr{E}$.

The goal of each RL algorithm $\alpha$ is to find a policy $\pi^* : \mathscr{E} \to \mathcal{A}$ which yields optimal expected returns. Such an algorithm $\alpha$ is viewed as a black box that takes as an input a trajectory set $\mathcal{D} \in \mathscr{E}^+$, where $\mathscr{E}^+$ is the ensemble of trajectory sets of undetermined size: $\mathscr{E}^+ = \bigcup_{T \in \mathbb{N}} \mathscr{E}^T$, and that outputs a policy $\pi_\mathcal{D}^\alpha$. Consequently, a RL algorithm is formalised as follows: $\alpha : \mathscr{E}^+ \to (\mathscr{E} \to \mathcal{A})$.

Such a high level definition of the RL algorithms allows to share trajectories between algorithms: a trajectory as a sequence of observations, actions, and rewards can be interpreted by any algorithm in its own decision process and state representation. For instance, RL algorithms classically rely on an MDP defined on a explicit or implicit state space representation $\mathcal{S}_\mathcal{D}^\alpha$ thanks to a projection $\Phi_\mathcal{D}^\alpha : \mathscr{E} \to \mathcal{S}_\mathcal{D}^\alpha$. Then, $\alpha$ trains its policy $\pi_{\mathcal{D}_T}^\alpha$ on the trajectories projected on its state space representation. Off-policy RL optimisation techniques compatible with this approach are numerous in

the literature (Watkins, 1989; Ernst et al., 2005; Mnih et al., 2013). As well, any post-treatment of the state set, any alternative decision process (Lovejoy, 1991), and any off-policy algorithm may be used. The algorithms are defined here as black boxes and the considered meta-algorithms will be indifferent to how the algorithms compute their policies, granted they satisfy the off-policy assumption.

## 2.2 ONLINE ALGORITHM SELECTION

The online learning approach is tackled in this article: different algorithms are experienced and evaluated during the data collection. Since it boils down to a classical exploration/exploitation trade-off, multi-armed bandit (Bubeck & Cesa-Bianchi, 2012) have been used for combinatorial search AS (Gagliolo & Schmidhuber, 2006; 2010) and evolutionary algorithm meta-learning (Fialho et al., 2010). The online AS problem for off-policy RL is novel and we define it as follows:

---

**Pseudo-code 1:** Online RL AS setting

**Data:** $\mathcal{D}_0 \leftarrow \emptyset$: trajectory set
**Data:** $\mathcal{P} \leftarrow \{\alpha^k\}_{k\in[\![1,K]\!]}$: algorithm portfolio
**Data:** $\mu : \mathscr{E} \to \mathbb{R}$: the objective function
**for** $\tau \leftarrow 1$ **to** $\infty$ **do**
    Select $\sigma(\mathcal{D}_{\tau-1}) = \sigma(\tau) \in \mathcal{P}$;
    Generate trajectory $\varepsilon_\tau$ with policy $\pi^{\sigma(\tau)}_{\mathcal{D}_{\tau-1}}$;
    Get return $\mu(\varepsilon_\tau)$;
    $\mathcal{D}_\tau \leftarrow \mathcal{D}_{\tau-1} \cup \{\varepsilon_\tau\}$;
**end**

---

- $\mathcal{D} \in \mathscr{E}^+$ is the current *trajectory set*;
- $\mathcal{P} = \{\alpha^k\}_{k\in[\![1,K]\!]}$ is the *portfolio* of off-policy RL algorithms;
- $\mu : \mathscr{E} \to \mathbb{R}$ is the *objective function*, generally set as the RL return.

Pseudo-code 1 formalises the online RL AS setting. A *meta-algorithm* is defined as a function from a trajectory set to the selection of an algorithm: $\sigma : \mathscr{E}^+ \to \mathcal{P}$. The meta-algorithm is queried at each meta-time $\tau = |\mathcal{D}_{\tau-1}|+1$, with input $\mathcal{D}_{\tau-1}$, and it ouputs algorithm $\sigma(\mathcal{D}_{\tau-1}) = \sigma(\tau) \in \mathcal{P}$ controlling with its policy $\pi^{\sigma(\tau)}_{\mathcal{D}_{\tau-1}}$ the generation of the trajectory $\varepsilon_\tau$ in the stochastic environment. Figure 2 illustrates the algorithm with a diagram flow. The final goal is to optimise the cumulative expected return. It is the expectation of the sum of rewards obtained after a run of $T$ trajectories:

$$\mathbb{E}_\sigma\left[\sum_{\tau=1}^{T} \mu(\varepsilon_\tau)\right] = \mathbb{E}_\sigma\left[\sum_{\tau=1}^{T} \mathbb{E}\mu^{\sigma(\tau)}_{\mathcal{D}^\sigma_{\tau-1}}\right], \tag{1}$$

with $\mathbb{E}\mu^\alpha_\mathcal{D} = \mathbb{E}_{\pi^\alpha_\mathcal{D}}[\mu(\varepsilon)]$ as a condensed notation for the expected return of policy $\pi^\alpha_\mathcal{D}$, trained on trajectory set $\mathcal{D}$ by algorithm $\alpha$. Equation 1 transforms the cumulative expected return into two nested

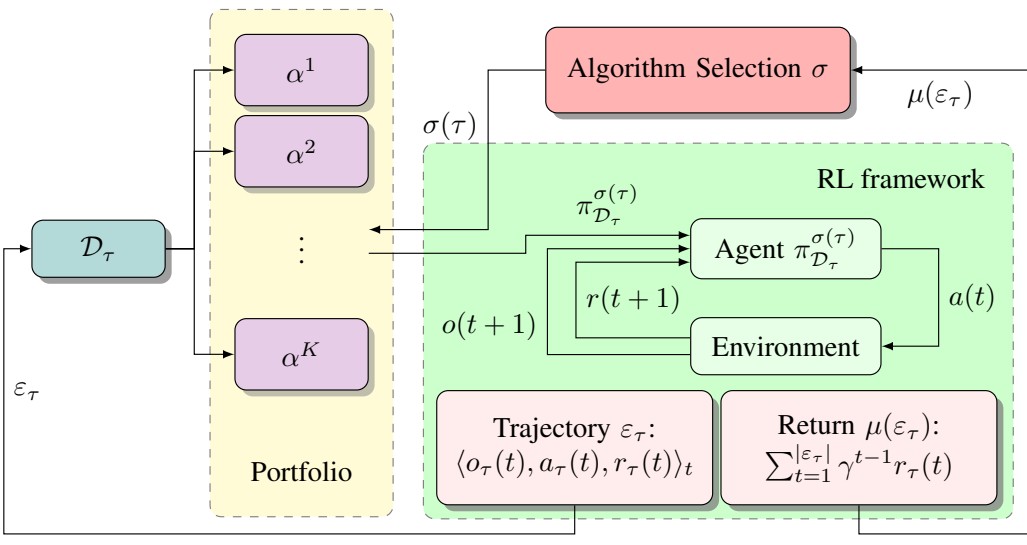

Figure 2: Algorithm selection for reinforcement learning flow diagram

expectations. The outside expectation $\mathbb{E}_\sigma$ assumes the meta-algorithm $\sigma$ fixed and averages over the trajectory set generation and the corresponding algorithms policies. The inside expectation $\mathbb{E}\mu$ assumes the policy fixed and averages over its possible trajectories in the stochastic environment. *Nota bene*: there are three levels of decision: meta-algorithm $\sigma$ selects algorithm $\alpha$ that computes policy $\pi$ that is in control. In this paper, the focus is at the meta-algorithm level.

## 2.3 META-ALGORITHM EVALUATION

In this paper, we focus on sample efficiency, where a sample is meant to be a trajectory. This is motivated by the following reasons. First, in most real-world systems, the major regret is on the task failure. The time expenditure is only a secondary concern that is already assessed by the discount factor dependency in the return. Second, it would be inconsistent to consider regret on a different time scale as the algorithm selection. Also, policy selection on non-episodic RL is known as a very difficult task where state-of-the-art algorithms only obtain regrets of the order of $\mathcal{O}(\sqrt{T}\log(T))$ on stationary policies Azar et al. (2013). Third, the regret on the decision steps cannot be assessed, since the rewards are discounted in the RL objective function. And finally, the bandit rewards (defined as the objective function in Section 2.2) may account for the length of the episode.

In order to evaluate the meta-algorithms, let us formulate two additional notations. First, the *optimal expected return* $\mathbb{E}\mu_\infty^*$ is defined as the highest expected return achievable by a policy of an algorithm in portfolio $\mathcal{P}$. Second, for every algorithm $\alpha$ in the portfolio, let us define $\sigma^\alpha$ as its *canonical meta-algorithm*, *i.e.* the meta-algorithm that always selects algorithm $\alpha$: $\forall\tau, \sigma^\alpha(\tau) = \alpha$. The *absolute pseudo-regret* $\overline{\rho}_{abs}^\sigma(T)$ defines the regret as the loss for not having controlled the trajectory with an optimal policy:

$$\overline{\rho}_{abs}^\sigma(T) = T\mathbb{E}\mu_\infty^* - \mathbb{E}_\sigma\left[\sum_{\tau=1}^T \mathbb{E}\mu_{\mathcal{D}_{\tau-1}^\sigma}^{\sigma(\tau)}\right]. \tag{2}$$

It is worth noting that an optimal meta-algorithm will unlikely yield a null regret because a large part of the absolute pseudo-regret is caused by the sub-optimality of the algorithm policies when the trajectory set is still of limited size. Indeed, the absolute pseudo-regret considers the regret for not selecting an optimal policy: it takes into account both the pseudo-regret of not selecting the best algorithm and the pseudo-regret of the algorithms for not finding an optimal policy. Since the meta-algorithm does not interfere with the training of policies, it ought not account for the pseudo-regret related to the latter.

## 2.4 RELATED WORK

Related to AS for RL, Schweighofer & Doya (2003) use meta-learning to tune a fixed RL algorithm in order to fit observed animal behaviour, which is a very different problem to ours. In Cauwet et al. (2014); Liu & Teytaud (2014), the RL AS problem is solved with a portfolio composed of online RL algorithms. The main limitation from these works relies on the fact that *on-policy* algorithms were used, which prevents them from sharing trajectories among algorithms (Cauwet et al., 2015). Meta-learning specifically for the eligibility trace parameter has also been studied in White & White (2016). Wang et al. (2016) study the learning process of RL algorithms and selects the best one for learning faster on a new task. This work is related to batch AS.

An intuitive way to solve the AS problem is to consider algorithms as arms in a multi-armed bandit setting. The bandit meta-algorithm selects the algorithm controlling the next trajectory $\varepsilon$ and the objective function $\mu(\varepsilon)$ constitutes the reward of the bandit. The aim of prediction with expert advice is to minimise the regret against the best expert of a set of predefined experts. When the experts learn during time, their performances evolve and hence the sequence of expert rewards is non-stationary.

The exponential weight algorithms (Auer et al., 2002b; Cesa-Bianchi & Lugosi, 2006) are designed for prediction with expert advice when the sequence of rewards of experts is generated by an oblivious adversary. This approach has been extended for competing against the best sequence of experts by adding in the update of weights a forgetting factor proportional to the mean reward (see Exp3.S in Auer et al. (2002b)), or by combining Exp3 with a concept drift detector Allesiardo & Féraud (2015).

The exponential weight algorithms have been extended to the case where the rewards are generated by any sequence of stochastic processes of unknown means (Besbes et al., 2014).

A recent article Graves et al. (2017) uses Exp3.S algorithm Auer et al. (2002b) for curriculum learning Bengio et al. (2009). The drawback of adversarial approaches is that they lead to very conservative algorithms which has to work against an adversary.

For handeling non-stationarity of rewards, another way is to assume that the rewards generated by each arm are not i.i.d., but are governed by some more complex stochastic processes. The stochastic bandit algorithm such as UCB can be extended to the case of switching bandits using a discount factor or a window to forget the past Garivier & Moulines (2011). Restless bandits Whittle (1988); Ortner et al. (2012) assume that a Markov chain governs the reward of arms independently of whether the learner is played or not the arm. These classes of bandit algorithms are not designed for experts that learn and hence evolve at each time step.

Our approach takes the opposite view of adversarial bandits: we design a stochastic algorithm specifically for curriculum learning based on the *doubling trick*. This reduction of the algorithm selection problem into several stochastic bandit problems with doubling time horizon begins to favour fast algorithms, and then more efficient algorithms.

## 3 EPOCHAL STOCHASTIC BANDIT

**ESBAS description –** To solve the off-policy RL AS problem, we propose a novel meta-algorithm called Epochal Stochastic Bandit AS (ESBAS). Because of the non-stationarity induced by the algorithm learning, the stochastic bandit cannot directly select algorithms. Instead, the stochastic bandit can choose fixed policies. To comply with this constraint, the meta-time scale is divided into epochs inside which the algorithms policies cannot be updated: the algorithms optimise their policies only when epochs start, in such a way that the policies are constant inside each epoch. This can be seen as a doubling trick. As a consequence and since the returns are bounded, at each new epoch, the problem can rigorously be cast into an independent stochastic $K$-armed bandit $\Xi$, with $K = |\mathcal{P}|$.

The ESBAS meta-algorithm is formally sketched in Pseudo-code 2 embedding UCB1 Auer et al. (2002a) as the stochastic $K$-armed bandit $\Xi$. The meta-algorithm takes as an input the set of algorithms in the portfolio. Meta-time scale is fragmented into epochs of exponential size. The $\beta^{\text{th}}$ epoch lasts $2^\beta$ meta-time steps, so that, at meta-time $\tau = 2^\beta$, epoch $\beta$ starts. At the beginning of each epoch, the ESBAS meta-algorithm asks each algorithm in the portfolio to update their current policy. Inside an epoch, the policy is never updated anymore. At the beginning of each epoch, a new $\Xi$ instance is reset and run. During the whole epoch, $\Xi$ selects at each meta-time step the algorithm in control of the next trajectory.

---

**Pseudo-code 2:** ESBAS with UCB1

**Data:** $\mathcal{D}_0, \mathcal{P}, \mu$: the online RL AS setting

**for** $\beta \leftarrow 0$ **to** $\infty$ **do**

    **for** $\alpha^k \in \mathcal{P}$ **do**

        $\pi^k_{\mathcal{D}_{2^\beta-1}}$: policy learnt by $\alpha^k$ on $\mathcal{D}_{2^\beta-1}$

    **end**

    $n \leftarrow 0, \forall \alpha^k \in \mathcal{P}, n^k \leftarrow 0$, and $x^k \leftarrow 0$

    **for** $\tau \leftarrow 2^\beta$ **to** $2^{\beta+1} - 1$ **do**

        $\alpha^{k_{max}} = \underset{\alpha^k \in \mathcal{P}}{\operatorname{argmax}} \left( x^k + \sqrt{\xi \frac{\log(n)}{n^k}} \right)$

        Generate trajectory $\varepsilon_\tau$ with policy $\pi^{k_{max}}_{\mathcal{D}_{2^\beta-1}}$

        Get return $\mu(\varepsilon_\tau)$, $\mathcal{D}_\tau \leftarrow \mathcal{D}_{\tau-1} \cup \{\varepsilon_\tau\}$

        $x^{k_{max}} \leftarrow \dfrac{n^{k_{max}} x^{k_{max}} + \mu(\varepsilon_\tau)}{n^{k_{max}} + 1}$

        $n^{k_{max}} \leftarrow n^{k_{max}} + 1$ and $n \leftarrow n + 1$

    **end**

**end**

---

**Theoretical analysis –** ESBAS intends to minimise the regret for not choosing the algorithm yielding the maximal return at a given meta-time $\tau$. It is short-sighted: it does not intend to optimise the algorithms learning. We define the short-sighted pseudo-regret as follows:

$$\overline{\rho}^\sigma_{ss}(T) = \mathbb{E}_\sigma \left[ \sum_{\tau=1}^{T} \left( \max_{\alpha \in \mathcal{P}} \mathbb{E}\mu^\alpha_{\mathcal{D}^\sigma_{\tau-1}} - \mathbb{E}\mu^{\sigma(\tau)}_{\mathcal{D}^\sigma_{\tau-1}} \right) \right]. \tag{3}$$

The short-sighted pseudo-regret depends on the *gaps* $\Delta_\beta^\alpha$: the difference of expected return between the best algorithm during epoch $\beta$ and algorithm $\alpha$. The smallest non null gap at epoch $\beta$ is noted $\Delta_\beta^\dagger$. We write its limit when $\beta$ tends to infinity with $\Delta_\infty^\dagger$.

Analysis relies on three assumptions that are formalised in Section B of the supplementary material. First, *more data is better data* states that algorithms improve on average from having additional data. Second, *order compatibility* assumes that, if a dataset enables to generate a better policy than another dataset, then, on average, adding new samples to both datasets should not change the dataset ordering. Third and last, let us introduce and discuss more in depth the *learning is fair* assumption. The *fairness of budget distribution* has been formalised in Cauwet et al. (2015). It is the property stating that every algorithm in the portfolio has as much resources as the others, in terms of computational time and data. It is an issue in most online AS problems, since the algorithm that has been the most selected has the most data, and therefore must be the most advanced one. A way to circumvent this issue is to select them equally, but, in an online setting, the goal of AS is precisely to select the best algorithm as often as possible. Our answer is to require that all algorithms in the portfolio are learning *off-policy*, *i.e.* without bias induced by the behavioural policy used in the learning dataset. By assuming that all algorithms learn off-policy, we allow *information sharing* Cauwet et al. (2015) between algorithms. They share the trajectories they generate. As a consequence, we can assume that every algorithm, the least or the most selected ones, will learn from the same trajectory set. Therefore, the control unbalance does not directly lead to unfairness in algorithms performances: all algorithms learn equally from all trajectories. However, unbalance might still remain in the exploration strategy if, for instance, an algorithm takes more benefit from the exploration it has chosen than the one chosen by another algorithm. For analysis purposes, we assumes the complete fairness of AS.

Based on those assumptions, three theorems show that ESBAS absolute pseudo-regret can be expressed in function of the absolute pseudo-regret of the best canonical algorithm and ESBAS short-sighted pseudo-regret. They also provide upper bounds on the ESBAS short-sighted pseudo-regret as a function of the order of magnitude of the gap $\Delta_\beta^\dagger$. Indeed, the stochastic multi-armed bandit algorithms have bounds that are, counter-intuitively, inversely proportional to the gaps between the best arm and the other ones. In particular if $\Delta_\beta^\dagger$ tends to 0, the algorithm selection might prove to be difficult, depending on the order of magnitude of it tending to 0. The full theoretical analysis can be found in the supplementary material, Section B. We provide here an intuitive overlook of its results. Table 1 numerically reports those bounds for a two-fold portfolio, depending on the nature of the algorithms. It must be read by line. According to the first column: the order of magnitude of $\Delta_\beta^\dagger$, the ESBAS short-sighted pseudo-regret bounds are displayed in the second column, and the third and fourth columns display the ESBAS absolute pseudo-regret bounds also depending on the order of magnitude of the best canonical algorithm absolute pseudo-regret: $\overline{\rho}_{abs}^{\sigma^*}(T)$.

Regarding the short-sighted upper bounds, the main result appears in the last line, when the algorithms converge to policies with different performance: ESBAS converges with a regret in $\mathcal{O}\left(\log^2(T)/\Delta_\infty^\dagger\right)$. Also, one should notice that the bounds of the first two lines are obtained by summing the gaps: this means that the algorithms are perceived equally good and that their gap goes beyond the threshold of distinguishability. This threshold is structurally at $\Delta_\beta^\dagger \in \mathcal{O}(1/\sqrt{T})$. The impossibility to determine which is the better algorithm is interpreted in Cauwet et al. (2014) as a budget issue. The meta-time necessary to distinguish through evaluation arms that are $\Delta_\beta^\dagger$ apart takes $\Theta(1/\Delta_\beta^{\dagger 2})$ meta-time steps. If the budget is inferior, then we are under the distinguishability threshold and the best bounds are obtained by summing up the gaps. As a consequence, if $\Delta_\beta^\dagger \in \mathcal{O}(1/\sqrt{T})$, then $1/\Delta_\beta^{\dagger 2} \in \Omega(T)$. However, the budget, *i.e.* the length of epoch $\beta$ starting at meta-time $T = 2^\beta$, equals $T$. $\Delta_\beta^\dagger \in \mathcal{O}(1/\sqrt{T})$ can therefore be considered as the structural limit of distinguishability between the algorithms.

Additionally, the absolute upper bounds are logarithmic in the best case and still inferior to $\mathcal{O}(\sqrt{T})$ in the worst case, which compares favorably with those of discounted UCB and Exp3.S in $\mathcal{O}(\sqrt{T\log(T)})$ and Rexp3 in $\mathcal{O}(T^{2/3})$, or the RL with Policy Advice's regret bounds of $\mathcal{O}(\sqrt{T}\log(T))$ on stationary policies Azar et al. (2013) (on non-episodic RL tasks).

Table 1: Bounds on $\overline{\rho}_{ss}^{\sigma^{\text{ESBAS}}}(T)$ and $\overline{\rho}_{abs}^{\sigma^{\text{ESBAS}}}(T)$ given various settings for a two-fold portfolio AS.

| $\Delta_\beta^\dagger$ | $\overline{\rho}_{ss}^{\sigma^{\text{ESBAS}}}(T)$ | $\overline{\rho}_{abs}^{\sigma^{\text{ESBAS}}}(T)$ in function of $\overline{\rho}_{abs}^{\sigma^*}(T)$ | |
|---|---|---|---|
| | | $\overline{\rho}_{abs}^{\sigma^*}(T) \in \mathcal{O}(\log(T))$ | $\overline{\rho}_{abs}^{\sigma^*}(T) \in \mathcal{O}(T^{1-c^*})$ |
| $\Theta(1/T)$ | $\mathcal{O}(\log(T))$ | $\mathcal{O}(\log(T))$ | |
| $\Theta(T^{-c^\dagger})$, and $c^\dagger \geq 0.5$ | $\mathcal{O}(T^{1-c^\dagger})$ | $\mathcal{O}(T^{1-c^\dagger})$ | $\mathcal{O}(T^{1-c^\dagger})$ |
| $\Theta(T^{-c^\dagger})$, and $c^\dagger < 0.5$ | $\mathcal{O}(T^{c^\dagger}\log(T))$ | $\mathcal{O}(T^{c^\dagger}\log(T))$ | $\mathcal{O}(T^{1-c^*})$, if $c^\dagger < 1 - c^*$ 
 $\mathcal{O}(T^{c^\dagger}\log(T))$, if $c^\dagger \geq 1 - c^*$ |
| $\Theta(1)$ | $\mathcal{O}\left(\log^2(T)/\Delta_\infty^\dagger\right)$ | $\mathcal{O}\left(\log^2(T)/\Delta_\infty^\dagger\right)$ | $\mathcal{O}(T^{1-c^*})$ |

# 4 ESBAS DIALOGUE EXPERIMENTS

ESBAS is particularly designed for RL tasks when it is impossible to update the policy after every transition or episode. Policy update is very costly in most real-world applications, such as dialogue systems (Khouzaimi et al., 2016) for which a growing batch setting is preferred (Lange et al., 2012). ESBAS practical efficiency is therefore illustrated on a dialogue negotiation game (Laroche & Genevay, 2016) that involves two players: the system $p_s$ and a user $p_u$. Their goal is to find an agreement among $4$ alternative options. At each dialogue, for each option $\eta$, players have a private uniformly drawn cost $\nu_\eta^p \sim \mathcal{U}[0, 1]$ to agree on it. Each player is considered fully empathetic to the other one. The details of the experiment can be found in the supplementary material, Section C.1.1.

All learning algorithms are using Fitted-$Q$ Iteration (Ernst et al., 2005), with a linear parametrisation and an $\epsilon_\beta$-greedy exploration : $\epsilon_\beta = 0.6^\beta$, $\beta$ being the epoch number. Several algorithms differing by their state space representation $\Phi^\alpha$ are considered: *simple*, *fast*, *simple-2*, *fast-2*, *n-$\zeta$-{simple/fast/simple-2/fast-2}*, and *constant-$\mu$*. See Section C.1.2 for their full descriptions.

The algorithms and ESBAS are playing with a stationary user simulator built through Imitation Learning from real-human data. All the results are averaged over 1000 runs. The performance figures plot the curves of algorithms individual performance $\sigma^\alpha$ against the ESBAS portfolio control $\sigma^{\text{ESBAS}}$ in function of the epoch (the scale is therefore logarithmic in meta-time). The performance is the average return of the RL problem. The ratio figures plot the average algorithm selection proportions of ESBAS at each epoch. We define the *relative pseudo regret* as the difference between the ESBAS absolute pseudo-regret and the absolute pseudo-regret of the best canonical meta-algorithm. Relative pseudo-regrets have a 95% confidence interval about $\pm 6 \approx \pm 1.5 \times 10^{-4}$ per trajectory. Extensive numerical results are provided in Table 2 of the supplementary material.

Figures 3a and 3b plot the typical curves obtained with ESBAS selecting from a portfolio of two learning algorithms. On Figure 3a, the ESBAS curve tends to reach more or less the best algorithm in each point as expected. Surprisingly, Figure 3b reveals that the algorithm selection ratios are not very strong in favour of one or another at any time. Indeed, the variance in trajectory set collection makes *simple* better on some runs until the end. ESBAS proves to be efficient at selecting the best algorithm for each run and unexpectedly obtains a negative relative pseudo-regret of -90. Figures 3c and 3d plot the typical curves obtained with ESBAS selecting from a portfolio constituted of a learning algorithm and an algorithm with a deterministic and stationary policy. ESBAS succeeds in remaining close to the best algorithm at each epoch and saves 5361 return value for not selecting the constant algorithm, but overall yields a regret for not using only the best algorithm. ESBAS also performs well on larger portfolios of 8 learners (see Figure 3e) with negative relative pseudo-regrets: $-10$, even if the algorithms are, on average, almost selected uniformly as Figure 3f reveals. Each individual run may present different ratios, depending on the quality of the trained policies. ESBAS also offers some curriculum learning, but more importantly, early bad policies are avoided.

Algorithms with a constant policy do not improve over time and the full reset of the $K$-multi armed bandit urges ESBAS to unnecessarily explore again and again the same underachieving algorithm. One easy way to circumvent this drawback is to use this knowledge and to not reset their arms. By operating this way, when the learning algorithm(s) start(s) outperforming the constant one, ESBAS

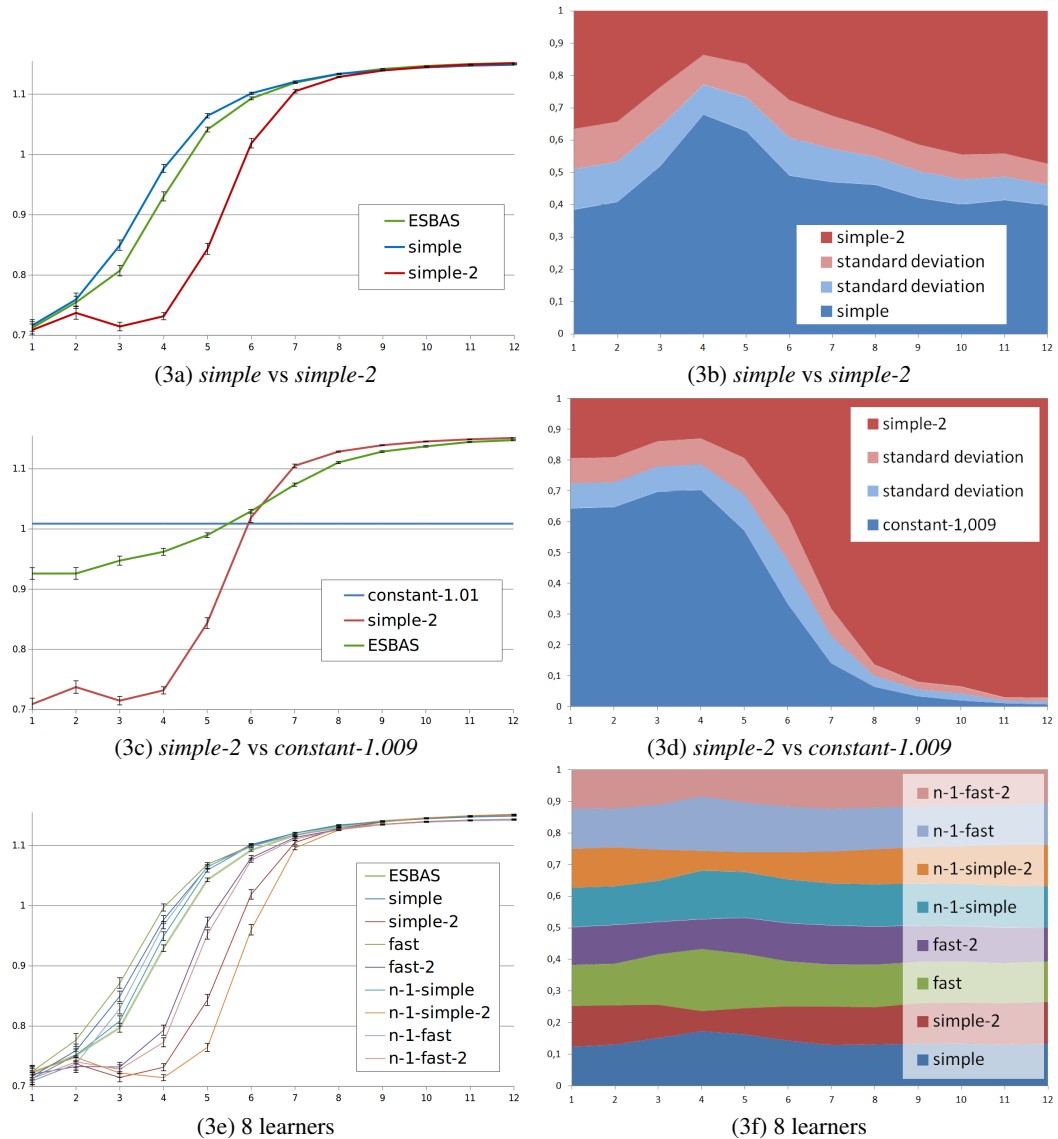

Figure 3: The figures on the top plot the performance over time. The figures on the bottom show the ESBAS selection ratios over the epochs.

simply neither exploits nor explores the constant algorithm anymore. Without arm reset for constant algorithms, ESBAS's learning curve follows perfectly the learning algorithm's learning curve when this one outperforms the constant algorithm and achieves strong negative relative pseudo-regrets. Again, the interested reader may refer to Table 2 in supplementary material for the numerical results. Still, another harmful phenomenon may happen: the constant algorithm overrides the natural exploration of the learning algorithm in the early stages, and when the learning algorithm finally outperforms the constant algorithm, its exploration parameter is already low. This can be observed in experiments with constant algorithm of expected return inferior to 1, as reported in Table 2.

## 5   SLIDING STOCHASTIC BANDIT ALGORITHM SELECTION

We propose to adapt ESBAS to a true online setting where algorithms update their policies after each transition. The stochastic bandit is now trained on a sliding window containing the last $\tau/2$ selections. Even though the arms are not stationary over this window, the bandit eventually forgets the oldest arm pulls. This algorithm is called SSBAS for Sliding Stochastic Bandit AS. Despite the lack of

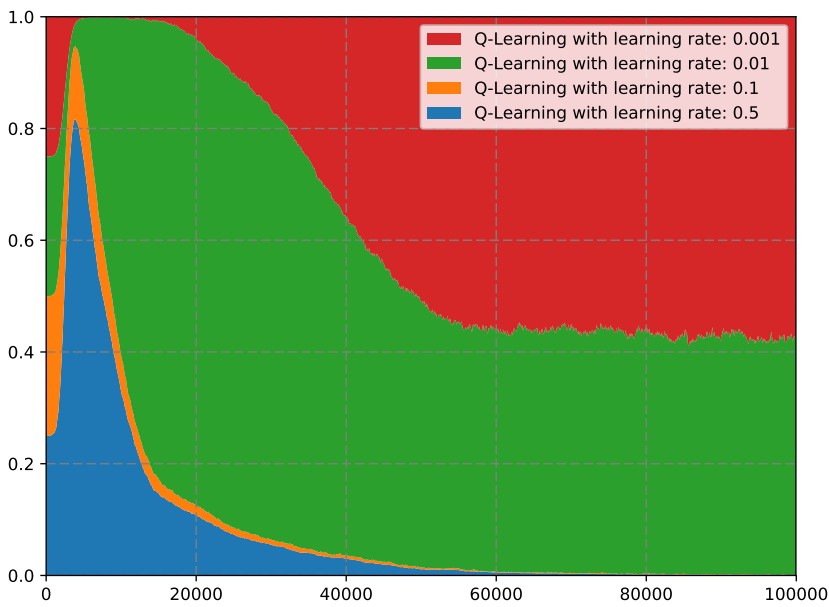

Figure 5: ratios (averaged over 3000 runs) obtained with SSBAS on the gridworld task.

theoretical convergence bounds, we demonstrate on two domains and two different meta-optimisation tasks that SSBAS impressively outperforming all algorithms in the portfolio by a wide margin.

## 5.1 GRIDWORLD DOMAIN

The goal here is to demonstrate that SSBAS can perform efficient hyper-parameter optimisation on a simple tabular domain: a 5x5 gridworld problem (see Figure 4), where the goal is to collect the fruits placed at each corner as fast as possible. The episodes terminate when all fruits have been collected or after 100 transitions. The objective function $\mu$ used to optimise the stochastic bandit $\Psi$ is no longer the RL return, but the time spent to collect all the fruits (200 in case of it did not). The agent has 18 possible positions and there are $2^4 - 1 = 15$ non-terminal fruits configurations, resulting in 270 states. The action set is $\mathcal{A} = \{N, E, S, W\}$. The reward function mean is 1 when eating a fruit, 0 otherwise. The reward function is corrupted with a strong Gaussian white noise of variance $\zeta^2 = 1$. The portfolio is composed of 4 $Q$-learning algorithms varying from each other by their learning rates: $\{0.001, 0.01, 0.1, 0.5\}$. They all have the same linearly annealing $\epsilon_\tau$-greedy exploration.

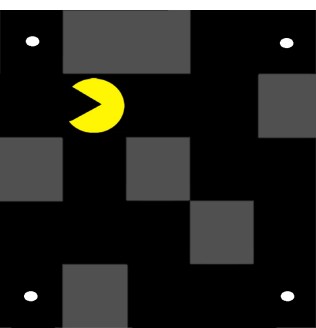

Figure 4: gridworld

The selection ratios displayed in 5 show that SSBAS selected the algorithm with the highest (0.5) learning rate in the first stages, enabling to propagate efficiently the reward signal through the visited states, then, over time preferentially chooses the algorithm with a learning rate of 0.01, which is less sensible to the reward noise, finally, SSBAS favours the algorithm with the finest learning rate (0.001). After 1 million episodes, SSBAS enables to save half a transition per episode on average as compared to the best fixed learning rate value (0.1), and two transitions against the worst fixed learning rate in the portfolio (0.001).

We compare SSBAS to the efficiency of a linearly annealing learning rate: $1/(1 + 0.0001\tau)$: SSBAS performs under 21 steps on average after $10^5$, while the linearly annealing learning rate algorithm still performs a bit over 21 steps after $10^6$ steps. This is because SSBAS can adapt the best performing learning rate over time.

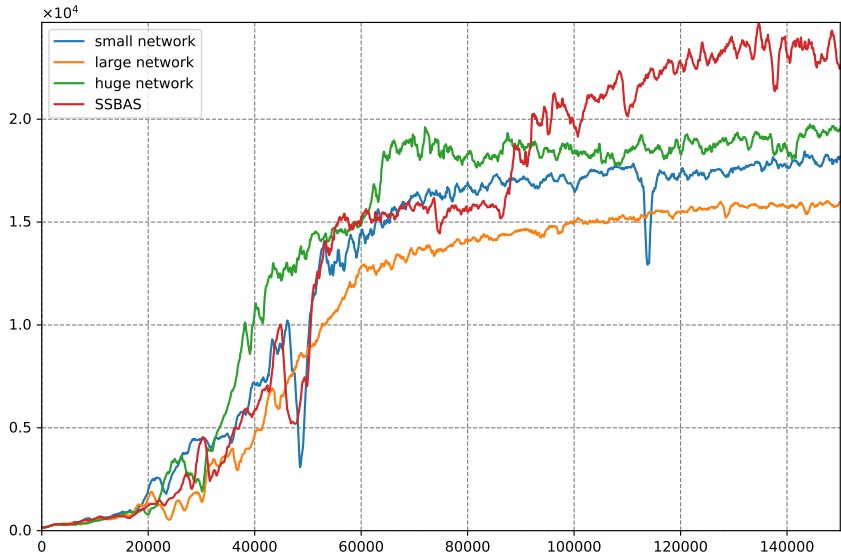

Figure 7: Comparative performance over time of SSBAS versus the algorithms in its portfolio on Q*bert (1 run).

We also compare SSBAS performance to Exp3.S's performance (Auer et al., 2002b). The analysis of the algorithm selection history shows that Exp3.S is too conservative and fails at efficiently select the shallowest algorithms in the beginning of the learning (number of steps at the 10000th episode: 28.3 for SSBAS vs 39.1 for Exp3.S), producing trajectories of lesser quality and therefore critically delaying the general training of all algorithms (number of steps at the 100000th episode: 20.9 for SSBAS vs 22.5 for Exp3.S). Overall, SSBAS outperforms Exp3.S by a significant and wide margin: number of steps averaged over all the training $10^5$ episodes: 28.7 for SSBAS vs 33.6 for Exp3.S.

## 5.2 ATARI DOMAIN: Q*BERT

We investigate here AS for deep RL on the Arcade Learning Environment (ALE, Bellemare et al. (2013)) and more precisely the game Q*bert (see a frame on Figure 6), where the goal is to step once on each block. Then a new similar level starts. In later levels, one needs to step twice on each block, and even later stepping again on the same blocks will cancel the colour change. We used three different settings of DQN instances: *small* uses the setting described in Mnih et al. (2013), *large* uses the setting in Mnih et al. (2015), and finally *huge* uses an even larger network (see the supplementary material, Section C.2 for details). DQN is known to reach a near-human level performance at Q*bert.

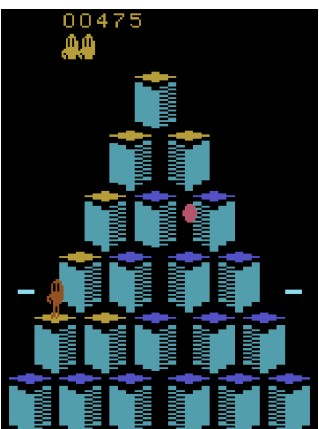

Figure 6: Q*bert

Our SSBAS instance runs 6 algorithms with 2 different random initialisations of each DQN setting. *Disclaimer:* contrarily to other experiments, each curve is the result of a single run, and the improvement might be aleatory. Indeed, the DQN training is very long and SSBAS needs to train all the models in parallel. A more computationally-efficient solution might be to use the same architecture as Osband et al. (2016).

7 reveals that SSBAS experiences a slight delay keeping in touch with the best setting performance during the initial learning phase, but, surprisingly, finds a better policy than the single algorithms in its portfolio and than the ones reported in the previous DQN articles. We observe that the *large* setting is surprisingly by far the worst one on the Q*bert task, implying the difficulty to predict which model is the most efficient for a new task. SSBAS allows to select online the best one.

## 6    CONCLUSION

In this article, we tackle the problem of selecting online off-policy RL algorithms. The problem is formalised as follows: from a fixed portfolio of algorithms, a meta-algorithm learns which one performs the best on the task at hand. Fairness of algorithm evaluation implies that the RL algorithms learn off-policy. ESBAS, a novel meta-algorithm, is proposed. Its principle is to divide the meta-time scale into epochs. Algorithms are allowed to update their policies only at the start each epoch. As the policies are constant inside each epoch, the problem can be cast into a stochastic multi-armed bandit. An implementation is detailed and a theoretical analysis leads to upper bounds on the regrets. ESBAS is designed for the growing batch RL setting. This limited online setting is required in many real-world applications where updating the policy requires a lot of resources.

Experiments are first led on a negotiation dialogue game, interacting with a human data-built simulated user. In most settings, not only ESBAS demonstrates its efficiency to select the best algorithm, but it also outperforms the best algorithm in the portfolio thanks to curriculum learning, and variance reduction similar to that of Ensemble Learning. Then, ESBAS is adapted to a full online setting, where algorithms are allowed to update after each transition. This meta-algorithm, called SSBAS, is empirically validated on a fruit collection task where it performs efficient hyper-parameter optimisation. SSBAS is also evaluated on the Q*bert Atari game, where it achieves a substantial improvement over the single algorithm counterparts.

We interpret ESBAS/SSBAS's success at reliably outperforming the best algorithm in the portfolio as the result of the four following potential added values. First, curriculum learning: ESBAS/SSBAS selects the algorithm that is the most fitted with the data size. This property allows for instance to use shallow algorithms when having only a few data and deep algorithms once collected a lot. Second, diversified policies: ESBAS/SSBAS computes and experiments several policies. Those diversified policies generate trajectories that are less redundant, and therefore more informational. As a result, the policies trained on these trajectories should be more efficient. Third, robustness: if one algorithm fails at finding good policies, it will soon be discarded. This property prevents the agent from repeating again and again the same obvious mistakes. Four and last, run adaptation: of course, there has to be an algorithm that is the best on average for one given task at one given meta-time. But depending on the variance in the trajectory collection, it did not necessarily train the best policy for each run. The ESBAS/SSBAS meta-algorithm tries and selects the algorithm that is the best at *each* run. Some of those properties are inherited by algorithm selection similarity with ensemble learning (Dieterich, 2002). Wiering & Van Hasselt (2008) uses a vote amongst the algorithms to decide the control of the next transition. Instead, ESBAS/SSBAS selects the best performing algorithm.

Regarding the portfolio design, it mostly depends on the available computational power per sample ratio. For practical implementations, we recommend to limit the use of two highly demanding algorithms, paired with several faster algorithms that can take care of first learning stages, and to use algorithms that are diverse regarding models, hypotheses, etc. Adding two algorithms that are too similar adds inertia, while they are likely to not be distinguishable by ESBAS/SSBAS. More detailed recommendations for building an efficient RL portfolio are left for future work.

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

# A GLOSSARY

| Symbol | Designation | First use |
|--------|-------------|-----------|
| $t$ | Reinforcement learning time *aka* RL-time | Section 2 |
| $\tau, T$ | Meta-algorithm time *aka* meta-time | Section 2 |
| $a(t)$ | Action taken at RL-time $t$ | Figure 1 |
| $o(t)$ | Observation made at RL-time $t$ | Figure 1 |
| $r(t)$ | Reward received at RL-time $t$ | Figure 1 |
| $\mathcal{A}$ | Action set | Section 2 |
| $\Omega$ | Observation set | Section 2 |
| $R_{min}$ | Lower bound of values taken by $R$ | Section 2 |
| $R_{max}$ | Upper bound of values taken by $R$ | Section 2 |
| $\varepsilon_\tau$ | Trajectory collected at meta-time $\tau$ | Section 2 |
| $|X|$ | Size of finite set/list/collection $X$ | Section 2 |
| $[\![a,b]\!]$ | Ensemble of integers comprised between $a$ and $b$ | Section 2 |
| $\mathscr{E}$ | Space of trajectories | Section 2 |
| $\gamma$ | Discount factor of the decision process | Section 2 |
| $\mu(\varepsilon_\tau)$ | Return of trajectory $\varepsilon_\tau$ *aka* objective function | Section 2 |
| $\mathcal{D}_T$ | Trajectory set collected until meta-time $T$ | Section 2 |
| $\varepsilon_\tau(t)$ | History of $\varepsilon_\tau$ until RL-time $t$ | Section 2 |
| $\pi$ | Policy | Section 2 |
| $\pi^*$ | Optimal policy | Section 2 |
| $\alpha$ | Algorithm | Section 2 |
| $\mathscr{E}^+$ | Ensemble of trajectory sets | Section 2 |
| $\mathcal{S}_\mathcal{D}^\alpha$ | State space of algorithm $\alpha$ from trajectory set $\mathcal{D}$ | Section 2 |
| $\Phi_\mathcal{D}^\alpha$ | State space projection of algorithm $\alpha$ from trajectory set $\mathcal{D}$ | Section 2 |
| $\pi_\mathcal{D}^\alpha$ | Policy learnt by algorithm $\alpha$ from trajectory set $\mathcal{D}$ | Section 2 |
| $\mathcal{P}$ | Algorithm set *aka* portfolio | Section 2.2 |
| $K$ | Size of the portfolio | Section 2.2 |
| $\sigma$ | Meta-algorithm | Section 2.2 |
| $\sigma(\tau)$ | Algorithm selected by meta-algorithm $\sigma$ at meta-time $\tau$ | Section 2.2 |
| $\mathbb{E}_{x_0}[f(x_0)]$ | Expected value of $f(x)$ conditionally to $x = x_0$ | Equation 1 |
| $\mathbb{E}\mu_\mathcal{D}^\alpha$ | Expected return of trajectories controlled by policy $\pi_\mathcal{D}^\alpha$ | Equation 1 |
| $\mathbb{E}\mu_\infty^*$ | Optimal expected return | Section 2.3 |
| $\sigma^\alpha$ | Canonical meta-algorithm exclusively selecting algorithm $\alpha$ | Section 2.3 |
| $\overline{\rho}_{abs}^\sigma(T)$ | Absolute pseudo-regret | Definition 1 |
| $\mathcal{O}(f(x))$ | Set of functions that get asymptotically dominated by $\kappa f(x)$ | Section 3 |
| $\kappa$ | Constant number | Theorem 3 |
| $\Xi$ | Stochastic $K$-armed bandit algorithm | Section 3 |
| $\beta$ | Epoch index | Section 3 |
| $\xi$ | Parameter of the UCB algorithm | Pseudo-code 2 |
| $\overline{\rho}_{ss}^\sigma(T)$ | Short-sighted pseudo-regret | Definition 2 |
| $\Delta$ | Gap between the best arm and another arm | Theorem 2 |
| $\dagger$ | Index of the second best algorithm | Theorem 2 |
| $\Delta_\beta^\dagger$ | Gap of the second best arm at epoch $\beta$ | Theorem 2 |
| $\lfloor x \rfloor$ | Rounding of $x$ at the closest integer below | Theorem 2 |
| $\sigma^{\text{ESBAS}}$ | The ESBAS meta-algorithm | Theorem 2 |
| $\Theta(f(x))$ | Set of functions asymptotically dominating $\kappa f(x)$ and dominated by $\kappa' f(x)$ | Table 1 |
| $\sigma^*$ | Best meta-algorithm among the canonical ones | Theorem 3 |

| Symbol | Designation | First use |
|---|---|---|
| $p, p_s, p_u$ | Player, system player, and (simulated) user player | Section C.1.1 |
| $\eta$ | Option to agree or disagree on | Section C.1.1 |
| $\nu_\eta^p$ | Cost of booking/selecting option $\nu$ for player $p$ | Section C.1.1 |
| $\mathcal{U}[a, b]$ | Uniform distribution between $a$ and $b$ | Section C.1.1 |
| $s_f$ | Final state reached in a trajectory | Section C.1.1 |
| $R^{p_s}(s_f)$ | Immediate reward received by the system player at the end of the dialogue | Section C.1.1 |
| REFPROP($\eta$) | Dialogue act consisting in proposing option $\eta$ | Section C.1.1 |
| ASKREPEAT | Dialogue act consisting in asking the other player to repeat what he said | Section C.1.1 |
| ACCEPT($\eta$) | Dialogue act consisting in accepting proposition $\eta$ | Section C.1.1 |
| ENDDIAL | Dialogue act consisting in ending the dialogue | Section C.1.1 |
| $SER_s^u$ | Sentence error rate of system $p_s$ listening to user $p_u$ | Section C.1.1 |
| $score_{asr}$ | Speech recognition score | Section C.1.1 |
| $\mathcal{N}(x, v)$ | Normal distribution of centre $x$ and variance $v^2$ | Section C.1.1 |
| REFINSIST | REFPROP($\eta$), with $\eta$ being the last proposed option | Section C.1.1 |
| REFNEWPROP | REFPROP($\eta$), with $\eta$ being the best option that has not been proposed yet | Section C.1.1 |
| ACCEPT | ACCEPT($\eta$), with $\eta$ being the last understood option proposition | Section C.1.1 |
| $\epsilon_\beta$ | $\epsilon$-greedy exploration in function of epoch $\beta$ | Section C.1.2 |
| $\Phi^\alpha$ | Set of features of algorithm $\alpha$ | Section C.1.2 |
| $\phi_0$ | Constant feature: always equal to 1 $\alpha$ | Section C.1.2 |
| $\phi_{asr}$ | ASR feature: equal to the last recognition score | Section C.1.2 |
| $\phi_{dif}$ | Cost feature: equal to the difference of cost of proposed and targeted options | Section C.1.2 |
| $\phi_t$ | RL-time feature | Section C.1.2 |
| $\phi_{noise}$ | Noise feature | Section C.1.2 |
| $simple$ | FQI with $\Phi = \{\phi_0, \phi_{asr}, \phi_{dif}, \phi_t\}$ | Section C.1.2 |
| $fast$ | FQI with $\Phi = \{\phi_0, \phi_{asr}, \phi_{dif}\}$ | Section C.1.2 |
| $simple\text{-}2$ | FQI with $\Phi = \{\phi_0, \phi_{asr}, \phi_{dif}, \phi_t, \phi_{asr}\phi_{dif}, \phi_t\phi_{asr}, \phi_{dif}\phi_t, \phi_{asr}^2, \phi_{dif}^2, \phi_t^2\}$ | Section C.1.2 |
| $fast\text{-}2$ | FQI with $\Phi = \{\phi_0, \phi_{asr}, \phi_{dif}, \phi_{asr}\phi_{dif}, \phi_{asr}^2, \phi_{dif}^2\}$ | Section C.1.2 |
| $n\text{-}1\text{-}simple$ | FQI with $\Phi = \{\phi_0, \phi_{asr}, \phi_{dif}, \phi_t, \phi_{noise}\}$ | Section C.1.2 |
| $n\text{-}1\text{-}fast$ | FQI with $\Phi = \{\phi_0, \phi_{asr}, \phi_{dif}, \phi_{noise}\}$ | Section C.1.2 |
| $n\text{-}1\text{-}simple\text{-}2$ | FQI with $\Phi = \{\phi_0, \phi_{asr}, \phi_{dif}, \phi_t, \phi_{noise}, \phi_{asr}\phi_{dif}, \phi_t\phi_{noise}, \phi_{asr}\phi_t,$ $\phi_{dif}\phi_{noise}, \phi_{asr}\phi_{noise}, \phi_{dif}\phi_t, \phi_{asr}^2, \phi_{dif}^2, \phi_t^2, \phi_{noise}^2\}$ | Section C.1.2 |
| $n\text{-}1\text{-}fast\text{-}2$ | FQI with $\Phi = \{\phi_0, \phi_{asr}, \phi_{dif}, \phi_t\}$ | Section C.1.2 |
| $constant\text{-}\mu$ | Non-learning algorithm with average performance $\mu$ | Section C.1.2 |
| $\zeta$ | Number of noisy features added to the feature set | Section C.1.2 |
| $\mathbb{P}(x\|y)$ | Probability that $X = x$ conditionally to $Y = y$ | Equation 35 |

# B    THEORETICAL ANALYSIS

The theoretical aspects of algorithm selection for reinforcement learning in general, and Epochal Stochastic Bandit Algorithm Selection in particular, are thoroughly detailed in this section. The proofs of the Theorems are provided in Sections E, F, and G. We recall and formalise the absolute pseudo-regret definition provided in Section 2.3.

**Definition 1** (Absolute pseudo-regret). *The absolute pseudo-regret $\overline{\rho}_{abs}^{\sigma}(T)$ compares the meta-algorithm's expected return with the optimal expected return:*

$$\overline{\rho}_{abs}^{\sigma}(T) = T\mathbb{E}\mu_{\infty}^{*} - \mathbb{E}_{\sigma}\left[\sum_{\tau=1}^{T}\mathbb{E}\mu_{\mathcal{D}_{\tau-1}^{\sigma}}^{\sigma(\tau)}\right]. \tag{4}$$

## B.1    ASSUMPTIONS

The theoretical analysis is hindered by the fact that AS, not only directly influences the return distribution, but also the trajectory set distribution and therefore the policies learnt by algorithms for next trajectories, which will indirectly affect the future expected returns. In order to allow policy comparison, based on relation on trajectory sets they are derived from, our analysis relies on two assumptions.

**Assumption 1** (More data is better data). *The algorithms train better policies with a larger trajectory set on average, whatever the algorithm that controlled the additional trajectory:*

$$\forall\mathcal{D}\in\mathscr{E}^{+}, \forall\alpha,\alpha'\in\mathcal{P}, \quad \mathbb{E}\mu_{\mathcal{D}}^{\alpha} \leq \mathbb{E}_{\alpha'}\left[\mathbb{E}\mu_{\mathcal{D}\cup\varepsilon^{\alpha'}}^{\alpha}\right]. \tag{5}$$

Assumption 1 states that algorithms are off-policy learners and that additional data cannot lead to performance degradation on average. An algorithm that is not off-policy could be biased by a specific behavioural policy and would therefore transgress this assumption.

**Assumption 2** (Order compatibility). *If an algorithm trains a better policy with one trajectory set than with another, then it remains the same, on average, after collecting an additional trajectory from any algorithm:*

$$\forall\mathcal{D},\mathcal{D}'\in\mathscr{E}^{+}, \ \forall\alpha,\alpha'\in\mathcal{P}, \quad \mathbb{E}\mu_{\mathcal{D}}^{\alpha} < \mathbb{E}\mu_{\mathcal{D}'}^{\alpha} \Rightarrow \mathbb{E}_{\alpha'}\left[\mathbb{E}\mu_{\mathcal{D}\cup\varepsilon^{\alpha'}}^{\alpha}\right] \leq \mathbb{E}_{\alpha'}\left[\mathbb{E}\mu_{\mathcal{D}'\cup\varepsilon^{\alpha'}}^{\alpha}\right]. \tag{6}$$

Assumption 2 states that a performance relation between two policies trained on two trajectory sets is preserved on average after adding another trajectory, whatever the behavioural policy used to generate it. From these two assumptions, Theorem 1 provides an upper bound in order of magnitude in function of the worst algorithm in the portfolio. It is verified for any meta-algorithm $\sigma$.

**Theorem 1** (Not worse than the worst). *The absolute pseudo-regret is bounded by the worst algorithm absolute pseudo-regret in order of magnitude:*

$$\forall\sigma, \quad \overline{\rho}_{abs}^{\sigma}(T) \in \mathcal{O}\left(\max_{\alpha\in\mathcal{P}}\overline{\rho}_{abs}^{\sigma^{\alpha}}(T)\right). \tag{7}$$

Contrarily to what the name of Theorem 1 suggests, a meta-algorithm might be worse than the worst algorithm (similarly, it can be better than the best algorithm), but not in order of magnitude. Its proof is rather complex for such an intuitive result because, in order to control all the possible outcomes, one needs to translate the selections of algorithm $\alpha$ with meta-algorithm $\sigma$ into the canonical meta-algorithm $\sigma^{\alpha}$'s view.

## B.2    SHORT-SIGHTED PSEUDO-REGRET ANALYSIS OF ESBAS

ESBAS intends to minimise the regret for not choosing the best algorithm at a given meta-time $\tau$. It is short-sighted: it does not intend to optimise the algorithms learning.

**Definition 2** (Short-sighted pseudo-regret). *The short-sighted pseudo-regret $\overline{\rho}_{ss}^{\sigma}(T)$ is the difference between the immediate best expected return algorithm and the one selected:*

$$\overline{\rho}_{ss}^{\sigma}(T) = \mathbb{E}_{\sigma}\left[\sum_{\tau=1}^{T}\left(\max_{\alpha\in\mathcal{P}}\mathbb{E}\mu_{\mathcal{D}_{\tau-1}^{\sigma}}^{\alpha} - \mathbb{E}\mu_{\mathcal{D}_{\tau-1}^{\sigma}}^{\sigma(\tau)}\right)\right]. \tag{8}$$

**Theorem 2** (ESBAS short-sighted pseudo-regret). *If the stochastic multi-armed bandit $\Xi$ guarantees a regret of order of magnitude $\mathcal{O}(\log(T)/\Delta_{\beta}^{\dagger})$, then:*

$$\overline{\rho}_{ss}^{\sigma^{\text{ESBAS}}}(T) \in \mathcal{O}\left(\sum_{\beta=0}^{\lfloor\log(T)\rfloor}\frac{\beta}{\Delta_{\beta}^{\dagger}}\right). \tag{9}$$

Theorem 2 expresses in order of magnitude an upper bound for the short-sighted pseudo-regret of ESBAS. But first, let define the *gaps*: $\Delta_{\beta}^{\alpha} = \max_{\alpha'\in\mathcal{P}}\mathbb{E}\mu_{\mathcal{D}_{2^{\beta}-1}^{\sigma^{\text{ESBAS}}}}^{\alpha'} - \mathbb{E}\mu_{\mathcal{D}_{2^{\beta}-1}^{\sigma^{\text{ESBAS}}}}^{\alpha}$. It is the difference of expected return between the best algorithm during epoch $\beta$ and algorithm $\alpha$. The smallest non null gap at epoch $\beta$ is noted: $\Delta_{\beta}^{\dagger} = \min_{\alpha\in\mathcal{P},\Delta_{\beta}^{\alpha}>0}\Delta_{\beta}^{\alpha}$. If $\Delta_{\beta}^{\dagger}$ does not exist, *i.e.* if there is no non-null gap, the regret is null.

Several upper bounds in order of magnitude on $\overline{\rho}_{ss}(T)$ can be easily deduced from Theorem 2, depending on an order of magnitude of $\Delta_{\beta}^{\dagger}$. See the corollaries in Section F.1, Table 1 and more generally Section 3 for a discussion.

### B.3 ESBAS ABSOLUTE PSEUDO-REGRET ANALYSIS

The short-sighted pseudo-regret optimality depends on the meta-algorithm itself. For instance, a poor deterministic algorithm might be optimal at meta-time $\tau$ but yield no new information, implying the same situation at meta-time $\tau + 1$, and so on. Thus, a meta-algorithm that exclusively selects the deterministic algorithm would achieve a short-sighted pseudo-regret equal to 0, but selecting other algorithms are, in the long run, more efficient. Theorem 2 is a necessary step towards the absolute pseudo-regret analysis.

The absolute pseudo-regret can be decomposed between the absolute pseudo-regret of the best canonical meta-algorithm (*i.e.* the algorithm that finds the best policy), the regret for not always selecting the best algorithm, and potentially not learning as fast, and the short-sighted regret: the regret for not gaining the returns granted by the best algorithm. This decomposition leads to Theorem 3 that provides an upper bound of the absolute pseudo-regret in function of the best canonical meta-algorithm, and the short-sighted pseudo-regret.

The *fairness of budget distribution* is the property stating that every algorithm in the portfolio has as much resources as the others, in terms of computational time and data. Section 3 discusses it at length. For analysis purposes, Theorem 3 assumes the fairness of AS:

**Assumption 3** (Learning is fair). *If one trajectory set is better than another for training one given algorithm, it is the same for other algorithms.*

$$\forall\alpha,\alpha'\in\mathcal{P}, \ \ \forall\mathcal{D},\mathcal{D}'\in\mathscr{E}^{+}, \ \ \ \mathbb{E}\mu_{\mathcal{D}}^{\alpha} < \mathbb{E}\mu_{\mathcal{D}'}^{\alpha} \Rightarrow \mathbb{E}\mu_{\mathcal{D}}^{\alpha'} \leq \mathbb{E}\mu_{\mathcal{D}'}^{\alpha'}. \tag{10}$$

**Theorem 3** (ESBAS absolute pseudo-regret upper bound). *Under assumption 3, if the stochastic multi-armed bandit $\Xi$ guarantees that the best arm has been selected in the $T$ first episodes at least $T/K$ times, with high probability $1 - \delta_T$, with $\delta_T \in \mathcal{O}(1/T)$, then:*

$$\exists\kappa > 0, \ \ \forall T \geq 9K^2, \ \ \ \overline{\rho}_{abs}^{\sigma^{\text{ESBAS}}}(T) \leq (3K+1)\overline{\rho}_{abs}^{\sigma^{*}}\left(\tfrac{T}{3K}\right) + \overline{\rho}_{ss}^{\sigma^{\text{ESBAS}}}(T) + \kappa\log(T), \tag{11}$$

*where meta-algorithm $\sigma^{*}$ selects exclusively algorithm $\alpha^{*} = \text{argmin}_{\alpha\in\mathcal{P}}\overline{\rho}_{abs}^{\sigma^{\alpha}}(T)$.*

Successive and Median Elimination (Even-Dar et al., 2002) and Upper Confidence Bound (Auer et al., 2002a) under some conditions (Audibert et al., 2010) are examples of appropriate $\Xi$ satisfying both conditions stated in Theorems 2 and 3. Again, see Table 1 and more generally Section 3 for a discussion of those bounds.

## C   Experimental details

### C.1   Dialogue experiments details

#### C.1.1   The Negotiation Dialogue Game

ESBAS practical efficiency is illustrated on a dialogue negotiation game (Laroche & Genevay, 2016) that involves two players: the system $p_s$ and a user $p_u$. Their goal is to find an agreement among 4 alternative options. At each dialogue, for each option $\eta$, players have a private uniformly drawn cost $\nu_\eta^p \sim \mathcal{U}[0, 1]$ to agree on it. Each player is considered fully empathetic to the other one. As a result, if the players come to an agreement, the system's immediate reward at the end of the dialogue is $R^{p_s}(s_f) = 2 - \nu_\eta^{p_s} - \nu_\eta^{p_u}$, where $s_f$ is the state reached by player $p_s$ at the end of the dialogue, and $\eta$ is the agreed option; if the players fail to agree, the final immediate reward is $R^{p_s}(s_f) = 0$, and finally, if one player misunderstands and agrees on a wrong option, the system gets the cost of selecting option $\eta$ without the reward of successfully reaching an agreement: $R^{p_s}(s_f) = -\nu_\eta^{p_s} - \nu_{\eta'}^{p_u}$.

Players act each one in turn, starting randomly by one or the other. They have four possible actions. First, REFPROP($\eta$): the player makes a proposition: option $\eta$. If there was any option previously proposed by the other player, the player refuses it. Second, ASKREPEAT: the player asks the other player to repeat its proposition. Third, ACCEPT($\eta$): the player accepts option $\eta$ that was understood to be proposed by the other player. This act ends the dialogue either way: whether the understood proposition was the right one or not. Four, ENDDIAL: the player does not want to negotiate anymore and ends the dialogue with a null reward.

Understanding through speech recognition of system $p_s$ is assumed to be noisy: with a sentence error rate of probability $SER_s^u = 0.3$, an error is made, and the system understands a random option instead of the one that was actually pronounced. In order to reflect human-machine dialogue asymmetry, the simulated user always understands what the system says: $SER_u^s = 0$. We adopt the way Khouzaimi et al. (2015) generate speech recognition confidence scores: $score_{asr} = \frac{1}{1+e^{-X}}$ where $X \sim \mathcal{N}(x, 0.2)$. If the player understood the right option $x = 1$, otherwise $x = 0$.

The system, and therefore the portfolio algorithms, have their action set restrained to five non parametric actions: REFINSIST $\Leftrightarrow$ REFPROP($\eta_{t-1}$), $\eta_{t-1}$ being the option lastly proposed by the system; REFNEWPROP $\Leftrightarrow$ REFPROP($\eta$), $\eta$ being the preferred one after $\eta_{t-1}$, ASKREPEAT, ACCEPT$\Leftrightarrow$ ACCEPT($\eta$), $\eta$ being the last understood option proposition and ENDDIAL.

#### C.1.2   Learning algorithms

All learning algorithms are using Fitted-$Q$ Iteration (Ernst et al., 2005), with a linear parametrisation and an $\epsilon_\beta$-greedy exploration : $\epsilon_\beta = 0.6^\beta$, $\beta$ being the epoch number. Six algorithms differing by their state space representation $\Phi^\alpha$ are considered:

- *simple*: state space representation of four features: the constant feature $\phi_0 = 1$, the last recognition score feature $\phi_{asr}$, the difference between the cost of the proposed option and the next best option $\phi_{dif}$, and finally an RL-time feature $\phi_t = \frac{0.1t}{0.1t+1}$. $\Phi^\alpha = \{\phi_0, \phi_{asr}, \phi_{dif}, \phi_t\}$.

- *fast*: $\Phi^\alpha = \{\phi_0, \phi_{asr}, \phi_{dif}\}$.

- *simple-2*: state space representation of ten second order polynomials of *simple* features. $\Phi^\alpha = \{\phi_0, \phi_{asr}, \phi_{dif}, \phi_t, \phi_{asr}^2, \phi_{dif}^2, \phi_t^2, \phi_{asr}\phi_{dif}, \phi_{asr}\phi_t, \phi_t\phi_{dif}\}$.

- *fast-2*: state space representation of six second order polynomials of *fast* features. $\Phi^\alpha = \{\phi_0, \phi_{asr}, \phi_{dif}, \phi_{asr}^2, \phi_{dif}^2, \phi_{asr}\phi_{dif}\}$.

- *n-ζ-{simple/fast/simple-2/fast-2}*: Versions of previous algorithms with $\zeta$ additional features of noise, randomly drawn from the uniform distribution in $[0, 1]$.

- *constant-μ*: the algorithm follows a deterministic policy of average performance $\mu$ without exploration nor learning. Those constant policies are generated with *simple-2* learning from a predefined batch of limited size.

### C.1.3 EVALUATION PROTOCOL

In all our experiments, ESBAS has been run with UCB parameter $\xi = 1/4$. We consider 12 epochs. The first and second epochs last 20 meta-time steps, then their lengths double at each new epoch, for a total of 40,920 meta-time steps and as many trajectories. $\gamma$ is set to 0.9. The algorithms and ESBAS are playing with a stationary user simulator built through Imitation Learning from real-human data. All the results are averaged over 1000 runs. The performance figures plot the curves of algorithms individual performance $\sigma^\alpha$ against the ESBAS portfolio control $\sigma^{\text{ESBAS}}$ in function of the epoch (the scale is therefore logarithmic in meta-time). The performance is the average return of the reinforcement learning return: it equals $\gamma^{|\epsilon|} R^{p_s}(s_f)$ in the negotiation game. The ratio figures plot the average algorithm selection proportions of ESBAS at each epoch. We define the *relative pseudo regret* as the difference between the ESBAS absolute pseudo-regret and the absolute pseudo-regret of the best canonical meta-algorithm. All relative pseudo-regrets, as well as the gain for not having chosen the worst algorithm in the portfolio, are provided in Table 2. Relative pseudo-regrets have a 95% confidence interval about $\pm 6 \approx \pm 1.5 \times 10^{-4}$ per trajectory.

### C.1.4 ASSUMPTIONS TRANSGRESSIONS

Several results show that, in practice, the assumptions are transgressed. Firstly, we observe that Assumption 3 is transgressed. Indeed, it states that if a trajectory set is better than another for a given algorithm, then it's the same for the other algorithms. Still, this assumption infringement does not seem to harm the experimental results. It even seems to help in general: while this assumption is consistent curriculum learning, it is inconsistent with the run adaptation property advanced in Subsection 6 that states that an algorithm might be the best on some run and another one on other runs.

And secondly, off-policy reinforcement learning algorithms exist, but in practice, we use state space representations that distort their off-policy property (Munos et al., 2016). However, experiments do not reveal any obvious bias related to the off/on-policiness of the trajectory set the algorithms train on.

### C.2 Q*BERT EXPERIMENT DETAILS

The three DQN networks (*small*, *large*, and *huge*) are built in a similar fashion, with relu activations at each layer except for the output layer which is linear, with RMSprop optimizer ($\rho = 0.95$ and $\epsilon = 10^{-7}$), and with He uniform initialisation. The hyperparameters used for training them are also the same and equal to the ones presented in the table hereinafter:

| hyperparameter | value |
|---|---|
| minibatch size | 32 |
| replay memory size | $1 \times 10^6$ |
| agent history length | 4 |
| target network update frequency | $5 \times 10^4$ |
| discount factor | 0.99 |
| action repeat | 20 |
| update frequency | 20 |
| learning rate | $2.5 \times 10^{-4}$ |
| exploration parameter $\epsilon$ | $5 \times t^{-1} \times 10^{-6}$ |
| replay start size | $5 \times 10^{-4}$ |
| no-op max | 30 |

Only their shapes differ:

- *small* has a first convolution layer with a 4x4 kernel and a 2x2 stride, and a second convolution layer with a 4x4 kernel and a 2x2 stride, followed by a dense layer of size 128, and finally the output layer is also dense.

- *large* has a first convolution layer with a 8x8 kernel and a 4x4 stride, and a second convolution layer with a 4x4 kernel and a 2x2 stride, followed by a dense layer of size 256, and finally the output layer is also dense.

- *huge* has a first convolution layer with a 8x8 kernel and a 4x4 stride, a second convolution layer with a 4x4 kernel and a 2x2 stride, and a third convolution layer with a 3x3 kernel and a 1x1 stride, followed by a dense layer of size 512, and finally the output layer is also dense.

# D   EXTENDED RESULTS OF THE DIALOGUE EXPERIMENTS

| Portfolio | w. best | w. worst |
|---|---:|---:|
| *simple-2 + fast-2* | 35 | -181 |
| *simple + n-1-simple-2* | -73 | -131 |
| *simple + n-1-simple* | 3 | -2 |
| *simple-2 + n-1-simple-2* | -12 | -38 |
| *all-4 + constant-1.10* | 21 | -2032 |
| *all-4 + constant-1.11* | -21 | -1414 |
| *all-4 + constant-1.13* | -10 | -561 |
| *all-4* | -28 | -275 |
| *all-2-simple + constant-1.08* | -41 | -2734 |
| *all-2-simple + constant-1.11* | -40 | -2013 |
| *all-2-simple + constant-1.13* | -123 | -799 |
| *all-2-simple* | -90 | -121 |
| *fast + simple-2* | -39 | -256 |
| *simple-2 + constant-1.01* | 169 | -5361 |
| *simple-2 + constant-1.11* | 53 | -1380 |
| *simple-2 + constant-1.11* | 57 | -1288 |
| *simple + constant-1.08* | 54 | -2622 |
| *simple + constant-1.10* | 88 | -1565 |
| *simple + constant-1.14* | -6 | -297 |
| *all-4 + all-4-n-1 + constant-1.09* | 25 | -2308 |
| *all-4 + all-4-n-1 + constant-1.11* | 20 | -1324 |
| *all-4 + all-4-n-1 + constant-1.14* | -16 | -348 |
| *all-4 + all-4-n-1* | -10 | -142 |
| *all-2-simple + all-2-n-1-simple* | -80 | -181 |
| *4*n-2-simple* | -20 | -20 |
| *4*n-3-simple* | -13 | -13 |
| *8*n-1-simple-2* | -22 | -22 |
| *simple-2 + constant-0.97* (no reset) | 113 | -7131 |
| *simple-2 + constant-1.05* (no reset) | 23 | -3756 |
| *simple-2 + constant-1.09* (no reset) | -19 | -2170 |
| *simple-2 + constant-1.13* (no reset) | -16 | -703 |
| *simple-2 + constant-1.14* (no reset) | -125 | -319 |

Table 2: ESBAS pseudo-regret after 12 epochs (*i.e.* 40,920 trajectories) compared with the best and the worst algorithms in the portfolio, in function of the algorithms in the portfolio (described in the first column). The '+' character is used to separate the algorithms. *all-4* means all the four learning algorithms described in Section C.1.2 *simple + fast + simple-2 + fast-2*. *all-4-n-1* means the same four algorithms with one additional feature of noise. Finally, *all-2-simple* means *simple + simple-2* and *all-2-n-1-simple* means *n-1-simple + n-1-simple-2*. On the second column, the redder the colour, the worse ESBAS is achieving in comparison with the best algorithm. Inversely, the greener the colour of the number, the better ESBAS is achieving in comparison with the best algorithm. If the number is neither red nor green, it means that the difference between the portfolio and the best algorithm is insignificant and that they are performing as good. This is already an achievement for ESBAS to be as good as the best. On the third column, the bluer the cell, the weaker is the worst algorithm in the portfolio. One can notice that positive regrets are always triggered by a very weak worst algorithm in the portfolio. In these cases, ESBAS did not allow to outperform the best algorithm in the portfolio, but it can still be credited with the fact it dismissed efficiently the very weak algorithms in the portfolio.

# E   NOT WORSE THAN THE WORST

**Theorem 1** (Not worse than the worst). *The absolute pseudo-regret is bounded by the worst algorithm absolute pseudo-regret in order of magnitude:*

$$\forall \sigma, \quad \overline{\rho}_{abs}^{\sigma}(T) \in \mathcal{O}\left(\max_{\alpha \in \mathcal{P}} \overline{\rho}_{abs}^{\sigma^{\alpha}}(T)\right). \tag{7}$$

*Proof.* From Definition 1:

$$\overline{\rho}_{abs}^{\sigma}(T) = T\mathbb{E}\mu_{\infty}^{*} - \mathbb{E}_{\sigma}\left[\sum_{\tau=1}^{T} \mathbb{E}\mu_{\mathcal{D}_{\tau-1}^{\sigma}}^{\sigma(\tau)}\right], \tag{12a}$$

$$\overline{\rho}_{abs}^{\sigma}(T) = T\mathbb{E}\mu_{\infty}^{*} - \sum_{\alpha \in \mathcal{P}} \mathbb{E}_{\sigma}\left[\sum_{i=1}^{|sub^{\alpha}(\mathcal{D}_{T}^{\sigma})|} \mathbb{E}\mu_{\mathcal{D}_{\tau_i^{\alpha}-1}^{\sigma}}^{\alpha}\right], \tag{12b}$$

$$\overline{\rho}_{abs}^{\sigma}(T) = \sum_{\alpha \in \mathcal{P}} \mathbb{E}_{\sigma}\left[|sub^{\alpha}(\mathcal{D}_{T}^{\sigma})|\mathbb{E}\mu_{\infty}^{*} - \sum_{i=1}^{|sub^{\alpha}(\mathcal{D}_{T}^{\sigma})|} \mathbb{E}\mu_{\mathcal{D}_{\tau_i^{\alpha}-1}^{\sigma}}^{\alpha}\right], \tag{12c}$$

where $sub^{\alpha}(\mathcal{D})$ is the subset of $\mathcal{D}$ with all the trajectories generated with algorithm $\alpha$, where $\tau_i^{\alpha}$ is the index of the $i^{\text{th}}$ trajectory generated with algorithm $\alpha$, and where $|S|$ is the cardinality of finite set $S$. By convention, let us state that $\mathbb{E}\mu_{\mathcal{D}_{\tau_i^{\alpha}-1}^{\sigma}}^{\alpha} = \mathbb{E}\mu_{\infty}^{*}$ if $|sub^{\alpha}(\mathcal{D}_{T}^{\sigma})| < i$. Then:

$$\overline{\rho}_{abs}^{\sigma}(T) = \sum_{\alpha \in \mathcal{P}} \sum_{i=1}^{T} \mathbb{E}_{\sigma}\left[\mathbb{E}\mu_{\infty}^{*} - \mathbb{E}\mu_{\mathcal{D}_{\tau_i^{\alpha}-1}^{\sigma}}^{\alpha}\right]. \tag{13}$$

To conclude, let us prove by mathematical induction the following inequality:

$$\mathbb{E}_{\sigma}\left[\mathbb{E}\mu_{\mathcal{D}_{\tau_i^{\alpha}-1}^{\sigma}}^{\alpha}\right] \geq \mathbb{E}_{\sigma^{\alpha}}\left[\mathbb{E}\mu_{\mathcal{D}_{i-1}^{\sigma^{\alpha}}}^{\alpha}\right]$$

is true by vacuity for $i = 0$: both left and right terms equal $\mathbb{E}\mu_{\emptyset}^{\alpha}$. Now let us assume the property true for $i$ and prove it for $i + 1$:

$$\mathbb{E}_{\sigma}\left[\mathbb{E}\mu_{\mathcal{D}_{\tau_{i+1}^{\alpha}-1}^{\sigma}}^{\alpha}\right] = \mathbb{E}_{\sigma}\left[\mathbb{E}\mu_{\mathcal{D}_{\tau_i^{\alpha}-1}^{\sigma} \cup \varepsilon^{\alpha} \cup \left(\mathcal{D}_{\tau_{i+1}^{\alpha}-1}^{\sigma} \setminus \mathcal{D}_{\tau_i^{\alpha}}^{\sigma}\right)}^{\alpha}\right], \tag{14a}$$

$$\mathbb{E}_{\sigma}\left[\mathbb{E}\mu_{\mathcal{D}_{\tau_{i+1}^{\alpha}-1}^{\sigma}}^{\alpha}\right] = \mathbb{E}_{\sigma}\left[\mathbb{E}\mu_{\mathcal{D}_{\tau_i^{\alpha}-1}^{\sigma} \cup \varepsilon^{\alpha} \cup \bigcup_{\tau=\tau_i^{\alpha}+1}^{\tau_{i+1}^{\alpha}-1} \varepsilon^{\sigma(\tau)}}^{\alpha}\right], \tag{14b}$$

$$\mathbb{E}_{\sigma}\left[\mathbb{E}\mu_{\mathcal{D}_{\tau_{i+1}^{\alpha}-1}^{\sigma}}^{\alpha}\right] = \mathbb{E}_{\sigma}\left[\mathbb{E}\mu_{\mathcal{D}_{\tau_i^{\alpha}-1}^{\sigma} \cup \varepsilon^{\alpha} \cup \bigcup_{\tau=1}^{\tau_{i+1}^{\alpha}-\tau_i^{\alpha}-1} \varepsilon^{\sigma(\tau_i^{\alpha}+\tau)}}^{\alpha}\right]. \tag{14c}$$

If $|sub^{\alpha}(\mathcal{D}_{T}^{\sigma})| \geq i+1$, by applying mathematical induction assumption, then by applying Assumption 2 and finally by applying Assumption 1 recursively, we infer that:

$$\mathbb{E}_{\sigma}\left[\mathbb{E}\mu_{\mathcal{D}_{\tau_i^{\alpha}-1}^{\sigma}}^{\alpha}\right] \geq \mathbb{E}_{\sigma^{\alpha}}\left[\mathbb{E}\mu_{\mathcal{D}_{i-1}^{\sigma^{\alpha}}}^{\alpha}\right], \tag{15a}$$

$$\mathbb{E}_{\sigma}\left[\mathbb{E}\mu_{\mathcal{D}_{\tau_i^{\alpha}-1}^{\sigma} \cup \varepsilon^{\alpha}}^{\alpha}\right] \geq \mathbb{E}_{\sigma^{\alpha}}\left[\mathbb{E}\mu_{\mathcal{D}_{i-1}^{\sigma^{\alpha}} \cup \varepsilon^{\alpha}}^{\alpha}\right], \tag{15b}$$

$$\mathbb{E}_{\sigma}\left[\mathbb{E}\mu^{\alpha}_{\mathcal{D}^{\sigma}_{\tau^{\alpha}_i-1}\cup\varepsilon^{\alpha}}\right] \geq \mathbb{E}_{\sigma^{\alpha}}\left[\mathbb{E}\mu^{\alpha}_{\mathcal{D}^{\sigma^{\alpha}}_i}\right], \tag{15c}$$

$$\mathbb{E}_{\sigma}\left[\mathbb{E}\mu^{\alpha}_{\mathcal{D}^{\sigma}_{\tau^{\alpha}_i-1}\cup\varepsilon^{\alpha}\cup\bigcup^{\tau^{\alpha}_{i+1}-\tau^{\alpha}_i-1}_{\tau=1}\varepsilon^{\sigma(\tau^{\alpha}_i+\tau)}}\right] \geq \mathbb{E}_{\sigma^{\alpha}}\left[\mathbb{E}\mu^{\alpha}_{\mathcal{D}^{\sigma^{\alpha}}_i}\right]. \tag{15d}$$

If $|sub^{\alpha}(\mathcal{D}^{\sigma}_T)| < i+1$, the same inequality is straightforwardly obtained, since, by convention $\mathbb{E}\mu^k_{\mathcal{D}^k_{\tau^k_{i+1}}} = \mathbb{E}\mu^*_{\infty}$, and since, by definition $\forall\mathcal{D}\in\mathscr{E}^+, \forall\alpha\in\mathcal{P}, \mathbb{E}\mu^*_{\infty}\geq\mathbb{E}\mu^{\alpha}_{\mathcal{D}}$.

The mathematical induction proof is complete. This result leads to the following inequalities:

$$\overline{\rho}^{\sigma}_{abs}(T) \leq \sum_{\alpha\in\mathcal{P}}\sum_{i=1}^{T}\mathbb{E}_{\sigma^{\alpha}}\left[\mathbb{E}\mu^*_{\infty} - \mathbb{E}\mu^{\alpha}_{\mathcal{D}^{\sigma^{\alpha}}_{i-1}}\right], \tag{16a}$$

$$\overline{\rho}^{\sigma}_{abs}(T) \leq \sum_{\alpha\in\mathcal{P}}\overline{\rho}^{\sigma^{\alpha}}_{abs}(T), \tag{16b}$$

$$\overline{\rho}^{\sigma}_{abs}(T) \leq K\max_{\alpha\in\mathcal{P}}\overline{\rho}^{\sigma^{\alpha}}_{abs}(T), \tag{16c}$$

which leads directly to the result:

$$\forall\sigma, \quad \overline{\rho}^{\sigma}_{abs}(T) \in \mathcal{O}\left(\max_{\alpha^k\in\mathcal{P}}\overline{\rho}^{\sigma^k}_{abs}(T)\right). \tag{17}$$

*This proof may seem to the reader rather complex for such an intuitive and loose result but algorithm selection $\sigma$ and the algorithms it selects may act tricky. For instance selecting algorithm $\alpha$ only when the collected trajectory sets contains misleading examples (i.e. with worse expected return than with an empty trajectory set) implies that the following unintuitive inequality is always true: $\mathbb{E}\mu^{\alpha}_{\mathcal{D}^{\sigma}_{\tau-1}} \leq \mathbb{E}\mu^{\alpha}_{\mathcal{D}^{\sigma^{\alpha}}_{\tau-1}}$. In order to control all the possible outcomes, one needs to translate the selections of algorithm $\alpha$ into $\sigma^{\alpha}$'s view.* □

## F  ESBAS SHORT-SIGHTED PSEUDO-REGRET UPPER ORDER OF MAGNITUDE

**Theorem 2** (ESBAS short-sighted pseudo-regret). *If the stochastic multi-armed bandit $\Xi$ guarantees a regret of order of magnitude $\mathcal{O}(\log(T)/\Delta_\beta^\dagger)$, then:*

$$\overline{\rho}_{ss}^{\sigma^{\text{ESBAS}}}(T) \in \mathcal{O}\left(\sum_{\beta=0}^{\lfloor \log(T) \rfloor} \frac{\beta}{\Delta_\beta^\dagger}\right). \tag{9}$$

*Proof.* By simplification of notation, $\mathbb{E}\mu_{\mathcal{D}_\beta}^\alpha = \mathbb{E}\mu_{\mathcal{D}_{2^\beta-1}^{\sigma^{\text{ESBAS}}}}^\alpha$. From Definition 2:

$$\overline{\rho}_{ss}^{\sigma^{\text{ESBAS}}}(T) = \mathbb{E}_{\sigma^{\text{ESBAS}}}\left[\sum_{\tau=1}^{T}\left(\max_{\alpha\in\mathcal{P}}\mathbb{E}\mu_{\mathcal{D}_{\tau-1}^{\sigma^{\text{ESBAS}}}}^\alpha - \mathbb{E}\mu_{\mathcal{D}_{\tau-1}^{\sigma^{\text{ESBAS}}}}^{\sigma^{\text{ESBAS}}(\tau)}\right)\right], \tag{18a}$$

$$\overline{\rho}_{ss}^{\sigma^{\text{ESBAS}}}(T) = \mathbb{E}_{\sigma^{\text{ESBAS}}}\left[\sum_{\tau=1}^{T}\left(\max_{\alpha\in\mathcal{P}}\mathbb{E}\mu_{\beta_\tau}^\alpha - \mathbb{E}\mu_{\beta_\tau}^{\sigma^{\text{ESBAS}}(\tau)}\right)\right], \tag{18b}$$

$$\overline{\rho}_{ss}^{\sigma^{\text{ESBAS}}}(T) \leq \mathbb{E}_{\sigma^{\text{ESBAS}}}\left[\sum_{\beta=0}^{\lfloor \log_2(T) \rfloor}\sum_{\tau=2^\beta}^{2^{\beta+1}-1}\left(\max_{\alpha\in\mathcal{P}}\mathbb{E}\mu_{\beta}^\alpha - \mathbb{E}\mu_{\beta}^{\sigma^{\text{ESBAS}}(\tau)}\right)\right], \tag{18c}$$

$$\overline{\rho}_{ss}^{\sigma^{\text{ESBAS}}}(T) \leq \sum_{\beta=0}^{\lfloor \log_2(T) \rfloor}\overline{\rho}_{ss}^{\sigma^{\text{ESBAS}}}(\beta), \tag{18d}$$

where $\beta_\tau$ is the epoch of meta-time $\tau$. A bound on short-sighted pseudo-regret $\overline{\rho}_{ss}^{\sigma^{\text{ESBAS}}}(\beta)$ for each epoch $\beta$ can then be obtained by the stochastic bandit $\Xi$ regret bounds in $\mathcal{O}\left(\log(T)/\Delta\right)$:

$$\overline{\rho}_{ss}^{\sigma^{\text{ESBAS}}}(\beta) = \mathbb{E}_{\sigma^{\text{ESBAS}}}\left[\sum_{\tau=2^\beta}^{2^{\beta+1}-1}\left(\max_{\alpha\in\mathcal{P}}\mathbb{E}\mu_{\beta}^\alpha - \mathbb{E}\mu_{\beta}^{\sigma^{\text{ESBAS}}(\tau)}\right)\right], \tag{19a}$$

$$\overline{\rho}_{ss}^{\sigma^{\text{ESBAS}}}(\beta) \in \mathcal{O}\left(\frac{\log(2^\beta)}{\Delta_\beta}\right), \tag{19b}$$

$$\overline{\rho}_{ss}^{\sigma^{\text{ESBAS}}}(\beta) \in \mathcal{O}\left(\frac{\beta}{\Delta_\beta}\right), \tag{19c}$$

$$\Leftrightarrow \exists\kappa_1 > 0, \qquad \overline{\rho}_{ss}^{\sigma^{\text{ESBAS}}}(\beta) \leq \frac{\kappa_1\beta}{\Delta_\beta}, \tag{19d}$$

$$\text{where} \qquad \frac{1}{\Delta_\beta} = \sum_{\alpha\in\mathcal{P}}\frac{1}{\Delta_\beta^\alpha} \tag{19e}$$

$$\text{and where} \qquad \Delta_\beta^\alpha = \begin{cases} +\infty & \text{if } \mathbb{E}\mu_\beta^\alpha = \max_{\alpha'\in\mathcal{P}}\mathbb{E}\mu_\beta^{\alpha'}, \\ \max_{\alpha'\in\mathcal{P}}\mathbb{E}\mu_\beta^{\alpha'} - \mathbb{E}\mu_\beta^\alpha & \text{otherwise.} \end{cases} \tag{19f}$$

Since we are interested in the order of magnitude, we can once again only consider the upper bound of $\frac{1}{\Delta_\beta}$:

$$\frac{1}{\Delta_\beta} \in \bigcup_{\alpha\in\mathcal{P}}\mathcal{O}\left(\frac{1}{\Delta_\beta^\alpha}\right), \tag{20a}$$

$$\frac{1}{\Delta_\beta} \in \mathcal{O}\left(\max_{\alpha \in \mathcal{P}} \frac{1}{\Delta_\beta^\alpha}\right), \tag{20b}$$

$$\Leftrightarrow \exists \kappa_2 > 0, \qquad \frac{1}{\Delta_\beta} \le \frac{\kappa_2}{\Delta_\beta^\dagger}, \tag{20c}$$

where the second best algorithm at epoch $\beta$ such that $\Delta_\beta^\dagger > 0$ is noted $\alpha_\beta^\dagger$. Injected in Equation 18d, it becomes:

$$\overline{\rho}_{ss}^{\sigma^{\text{ESBAS}}}(T) \le \kappa_1 \kappa_2 \sum_{\beta=0}^{\lfloor \log_2(T) \rfloor} \frac{\beta}{\Delta_\beta^\dagger}, \tag{21}$$

which proves the result. □

### F.1 Corollaries of Theorem 2

**Corollary 1.** *If* $\Delta_\beta^\dagger \in \Theta(1)$*, then* $\overline{\rho}_{ss}^{\sigma^{\text{ESBAS}}}(T) \in \mathcal{O}\left(\frac{\log^2(T)}{\Delta_\infty^\dagger}\right)$*, where* $\Delta_\infty^\dagger = \mu_\infty^* - \mu_\infty^\dagger > 0$*.*

*Proof.* $\Delta_\beta^\dagger \in \Omega(1)$ means that only one algorithm $\alpha^*$ converges to the optimal asymptotic performance $\mu_\infty^*$ and that $\exists \Delta_\infty^\dagger = \mu_\infty^* - \mu_\infty^\dagger > 0$ such that $\forall \epsilon_2 > 0$, $\exists \beta_1 \in \mathbb{N}$, such that $\forall \beta \ge \beta_1$, $\Delta_\beta^\dagger > \Delta_\infty^\dagger - \epsilon$. In this case, the following bound can be deduced from equation 21:

$$\overline{\rho}_{ss}^{\sigma^{\text{ESBAS}}}(T) \le \kappa_4 + \sum_{\beta=\beta_1}^{\lfloor \log(T) \rfloor} \frac{\kappa_1 \kappa_2}{\Delta_\infty^\dagger - \epsilon} \beta, \tag{22a}$$

$$\overline{\rho}_{ss}^{\sigma^{\text{ESBAS}}}(T) \le \kappa_4 + \frac{\kappa_1 \kappa_2 \log^2(T)}{2(\Delta_\infty^\dagger - \epsilon)}, \tag{22b}$$

where $\kappa_4$ is a constant equal to the short-sighted pseudo-regret before epoch $\beta_1$:

$$\kappa_4 = \overline{\rho}_{ss}^{\sigma^{\text{ESBAS}}}\left(2^{\beta_1 - 1}\right) \tag{23}$$

Equation 22b directly leads to the corollary. □

**Corollary 2.** *If* $\Delta_\beta^\dagger \in \Theta\left(\beta^{-m^\dagger}\right)$*, then* $\overline{\rho}_{ss}^{\sigma^{\text{ESBAS}}}(T) \in \mathcal{O}\left(\log^{m^\dagger+2}(T)\right)$*.*

*Proof.* If $\Delta_\beta^\dagger$ decreases slower than polynomially in epochs, which implies decreasing polylogarithmically in meta-time, *i.e.* $\exists \kappa_5 > 0, \exists m^\dagger > 0, \exists \beta_2 \in \mathbb{N}$, such that $\forall \beta \ge \beta_2, \Delta_\beta^\dagger > \kappa_5 \beta^{-m^\dagger}$, then, from Equation 21:

$$\overline{\rho}_{ss}^{\sigma^{\text{ESBAS}}}(T) \le \kappa_6 + \sum_{\beta=\beta_2}^{\lfloor \log(T) \rfloor} \frac{\kappa_1 \kappa_2}{\kappa_5 \beta^{-m^\dagger}} \beta, \tag{24a}$$

$$\overline{\rho}_{ss}^{\sigma^{\text{ESBAS}}}(T) \le \kappa_6 + \sum_{\beta=\beta_2}^{\lfloor \log(T) \rfloor} \frac{\kappa_1 \kappa_2}{\kappa_5} \beta^{m^\dagger+1}, \tag{24b}$$

$$\overline{\rho}_{ss}^{\sigma^{\text{ESBAS}}}(T) \le \frac{\kappa_1 \kappa_2}{\kappa_5} \log^{m^\dagger+2}(T), \tag{24c}$$

where $\kappa_6$ is a constant equal to the short-sighted pseudo-regret before epoch $\beta_2$:

$$\kappa_6 = \overline{\rho}_{ss}^{\sigma^{\text{ESBAS}}}\left(2^{\beta_2-1}\right). \tag{25}$$

Equation 24c directly leads to the corollary. $\qquad\square$

**Corollary 3.** *If $\Delta_\beta^\dagger \in \Theta\left(T^{-c^\dagger}\right)$, then $\overline{\rho}_{ss}^{\sigma^{\text{ESBAS}}}(T) \in \mathcal{O}\left(T^{c^\dagger}\log(T)\right)$.*

*Proof.* If $\Delta_\beta^\dagger$ decreases slower than a fractional power of meta-time $T$, then $\exists \kappa_7 > 0, 0 < c^\dagger < 1$, $\exists \beta_3 \in \mathbb{N}$, such that $\forall \beta \geq \beta_3, \Delta_\beta^\dagger > \kappa_7 T^{-c^\dagger}$, and therefore, from Equation 21:

$$\overline{\rho}_{ss}^{\sigma^{\text{ESBAS}}}(T) \leq \kappa_8 + \sum_{\beta=\beta_3}^{\lfloor \log(T)\rfloor} \frac{\kappa_1\kappa_2}{\kappa_7\tau^{-c^\dagger}}\beta, \tag{26a}$$

$$\overline{\rho}_{ss}^{\sigma^{\text{ESBAS}}}(T) \leq \kappa_8 + \sum_{\beta=\beta_3}^{\lfloor \log(T)\rfloor} \frac{\kappa_1\kappa_2}{\kappa_7\left(2^\beta\right)^{-c^\dagger}}\beta, \tag{26b}$$

$$\overline{\rho}_{ss}^{\sigma^{\text{ESBAS}}}(T) \leq \kappa_8 + \sum_{\beta=\beta_3}^{\lfloor \log(T)\rfloor} \frac{\kappa_1\kappa_2}{\kappa_7}\beta\left(2^{c^\dagger}\right)^\beta, \tag{26c}$$

where $\kappa_8$ is a constant equal to the short-sighted pseudo-regret before epoch $\beta_3$:

$$\kappa_8 = \overline{\rho}_{ss}^{\sigma^{\text{ESBAS}}}\left(2^{\beta_3-1}\right). \tag{27}$$

The sum in Equation 26c is solved as follows:

$$\sum_{i=i_0}^{n} ix^i = x\sum_{i=i_0}^{n} ix^{i-1}, \tag{28a}$$

$$\sum_{i=i_0}^{n} ix^i = x\sum_{i=i_0}^{n} \frac{d\left(x^i\right)}{dx}, \tag{28b}$$

$$\sum_{i=i_0}^{n} ix^i = x\frac{d\left(\sum_{i=i_0}^{n} x^i\right)}{dx}, \tag{28c}$$

$$\sum_{i=i_0}^{n} ix^i = x\frac{d\left(\frac{x^{n+1}-x^{i_0}}{x-1}\right)}{dx}, \tag{28d}$$

$$\sum_{i=i_0}^{n} ix^i = \frac{x}{(x-1)^2}\left((x-1)nx^n - x^n - (x-1)i_0 x^{i_0-1} + x^{i_0}\right). \tag{28e}$$

This result, injected in Equation 26c, induces that $\forall \epsilon_3 > 0, \exists T_1 \in \mathbb{N}, \forall T \geq T_1$:

$$\overline{\rho}_{ss}^{\sigma^{\text{ESBAS}}}(T) \leq \kappa_8 + \frac{\kappa_1\kappa_2(1+\epsilon')2^{c^\dagger}}{\kappa_7(2^{c^\dagger}-1)}\log(T)2^{c^\dagger\log(T)}, \tag{29a}$$

$$\overline{\rho}_{ss}^{\sigma^{\text{ESBAS}}}(T) \leq \kappa_8 + \frac{\kappa_1\kappa_2(1+\epsilon')2^{c^\dagger}}{\kappa_7(2^{c^\dagger}-1)}T^{c^\dagger}\log(T), \tag{29b}$$

which proves the corollary. $\qquad\square$

## G   ESBAS ABSOLUTE PSEUDO-REGRET BOUND

**Theorem 3** (ESBAS absolute pseudo-regret upper bound). *Under assumption 3, if the stochastic multi-armed bandit $\Xi$ guarantees that the best arm has been selected in the $T$ first episodes at least $T/K$ times, with high probability $1 - \delta_T$, with $\delta_T \in \mathcal{O}(1/T)$, then:*

$$\exists \kappa > 0, \ \ \forall T \geq 9K^2, \quad \overline{\rho}_{abs}^{\sigma^{\mathrm{ESBAS}}}(T) \leq (3K+1)\overline{\rho}_{abs}^{\sigma^*}\left(\frac{T}{3K}\right) + \overline{\rho}_{ss}^{\sigma^{\mathrm{ESBAS}}}(T) + \kappa \log(T), \quad (11)$$

*where meta-algorithm $\sigma^*$ selects exclusively algorithm $\alpha^* = \mathrm{argmin}_{\alpha \in \mathcal{P}} \, \overline{\rho}_{abs}^{\sigma^\alpha}(T)$.*

*Proof.* The ESBAS absolute pseudo-regret is written with the following notation simplifications :
$\mathcal{D}_{\tau-1} = \mathcal{D}_{\tau-1}^{\sigma^{\mathrm{ESBAS}}}$ and $k_\tau = \sigma^{\mathrm{ESBAS}}(\tau)$:

$$\overline{\rho}_{abs}^{\sigma^{\mathrm{ESBAS}}}(T) = T\mathbb{E}\mu_\infty^* - \mathbb{E}_{\sigma^{\mathrm{ESBAS}}}\left[\sum_{\tau=1}^T \mathbb{E}\mu_{\mathcal{D}_{\tau-1}^{\sigma^{\mathrm{ESBAS}}}}^{\sigma^{\mathrm{ESBAS}}(\tau)}\right], \quad (30a)$$

$$\overline{\rho}_{abs}^{\sigma^{\mathrm{ESBAS}}}(T) = T\mathbb{E}\mu_\infty^* - \mathbb{E}_{\sigma^{\mathrm{ESBAS}}}\left[\sum_{\tau=1}^T \mathbb{E}\mu_{\mathcal{D}_{\tau-1}}^{k_\tau}\right]. \quad (30b)$$

Let $\sigma^*$ denote the algorithm selection selecting exclusively $\alpha^*$, and $\alpha^*$ be the algorithm minimising the algorithm absolute pseudo-regret:

$$\alpha^* = \underset{\alpha^k \in \mathcal{P}}{\mathrm{argmin}} \, \overline{\rho}_{abs}^{\sigma^k}(T). \quad (31)$$

Note that $\sigma^*$ is the optimal constant algorithm selection at horizon $T$, but it is not necessarily the optimal algorithm selection: there might exist, and there probably exists a non constant algorithm selection yielding a smaller pseudo-regret.

The ESBAS absolute pseudo-regret $\overline{\rho}_{abs}^{\sigma^{\mathrm{ESBAS}}}(T)$ can be decomposed into the pseudo-regret for not having followed the optimal constant algorithm selection $\sigma^*$ and the pseudo-regret for not having selected the algorithm with the highest return, *i.e.* between the pseudo-regret on the trajectory and the pseudo-regret on the immediate optimal return:

$$\overline{\rho}_{abs}^{\sigma^{\mathrm{ESBAS}}}(T) = T\mathbb{E}\mu_\infty^* - \mathbb{E}_{\sigma^{\mathrm{ESBAS}}}\left[\sum_{\tau=1}^T \mathbb{E}\mu_{sub^*(\mathcal{D}_{\tau-1})}^*\right]$$
$$+ \mathbb{E}_{\sigma^{\mathrm{ESBAS}}}\left[\sum_{\tau=1}^T \mathbb{E}\mu_{sub^*(\mathcal{D}_{\tau-1})}^*\right] - \mathbb{E}_{\sigma^{\mathrm{ESBAS}}}\left[\sum_{\tau=1}^T \mathbb{E}\mu_{\mathcal{D}_{\tau-1}}^{k_\tau}\right], \quad (32)$$

where $\mathbb{E}\mu_{sub^*(\mathcal{D}_{\tau-1})}^*$ is the expected return of policy $\pi_{sub^*(\mathcal{D}_{\tau-1})}^*$, learnt by algorithm $\alpha^*$ on trajectory set $sub^*(\mathcal{D}_{\tau-1})$, which is the trajectory subset of $\mathcal{D}_{\tau-1}$ obtained by removing all trajectories that were not generated with algorithm $\alpha^*$.

First line of Equation 32 can be rewritten as follows:

$$T\mathbb{E}\mu_\infty^* - \mathbb{E}_{\sigma^{\mathrm{ESBAS}}}\left[\sum_{\tau=1}^T \mathbb{E}\mu_{sub^*(\mathcal{D}_{\tau-1})}^*\right] = \sum_{\tau=1}^T \left(\mathbb{E}\mu_\infty^* - \mathbb{E}_{\sigma^{\mathrm{ESBAS}}}\left[\mathbb{E}\mu_{sub^*(\mathcal{D}_{\tau-1})}^*\right]\right). \quad (33)$$

The key point in Equation 33 is to evaluate the size of $sub^*(\mathcal{D}_{\tau-1})$.

On the one side, Assumption 3 of fairness states that one algorithm learns as fast as any another over any history. The asymptotically optimal algorithm(s) when $\tau \to \infty$ is(are) therefore the same one(s)

whatever the the algorithm selection is. On the other side, let $1 - \delta_\tau$ denote the probability, that at time $\tau$, the following inequality is true:

$$|sub^*(\mathcal{D}_{\tau-1})| \geq \left\lfloor \frac{\tau-1}{3K} \right\rfloor. \tag{34}$$

With probability $\delta_\tau$, inequality 34 is not guaranteed and nothing can be inferred about $\mathbb{E}\mu^*_{sub^*(\mathcal{D}_{\tau-1})}$, except it is bounded under by $R_{min}/(1-\gamma)$. Let $\mathscr{E}^{\tau-1}_{3K}$ be the subset of $\mathscr{E}^{\tau-1}$ such that $\forall \mathcal{D} \in \mathscr{E}^{\tau-1}_{3K}, |sub^*(\mathcal{D})| \geq \lfloor (\tau-1)/3K \rfloor$. Then, $\delta_\tau$ can be expressed as follows:

$$\delta_\tau = \sum_{\mathcal{D} \in \mathscr{E}^{\tau-1} \setminus \mathscr{E}^{\tau-1}_{3K}} \mathbb{P}(\mathcal{D}|\sigma^{\text{ESBAS}}). \tag{35}$$

With these new notations:

$$\mathbb{E}\mu^*_\infty - \mathbb{E}_{\sigma^{\text{ESBAS}}}\left[\mathbb{E}\mu^*_{sub^*(\mathcal{D}_{\tau-1})}\right] \leq \mathbb{E}\mu^*_\infty - \sum_{\mathcal{D} \in \mathscr{E}^{\tau-1}_{3K}} \mathbb{P}(\mathcal{D}|\sigma^{\text{ESBAS}})\mathbb{E}\mu^*_{sub^*(\mathcal{D})} - \delta_\tau \frac{R_{min}}{1-\gamma}, \tag{36a}$$

$$\begin{aligned}
&\mathbb{E}\mu^*_\infty - \mathbb{E}_{\sigma^{\text{ESBAS}}}\left[\mathbb{E}\mu^*_{sub^*(\mathcal{D}_{\tau-1})}\right] \\
&\leq (1-\delta_\tau)\mathbb{E}\mu^*_\infty - \sum_{\mathcal{D} \in \mathscr{E}^{\tau-1}_{3K}} \mathbb{P}(\mathcal{D}|\sigma^{\text{ESBAS}})\mathbb{E}\mu^*_{sub^*(\mathcal{D})} + \delta_\tau \left( \mathbb{E}\mu^*_\infty - \frac{R_{min}}{1-\gamma} \right).
\end{aligned} \tag{36b}$$

Let consider $\mathscr{E}^*(\alpha, N)$ the set of all sets $\mathcal{D}$ such that $|sub^\alpha(\mathcal{D})| = N$ and such that last trajectory in $\mathcal{D}$ was generated by $\alpha$. Since ESBAS, with $\Xi$, a stochastic bandit with regret in $\mathcal{O}(\log(T)/\Delta)$, guarantees that all algorithms will eventually be selected an infinity of times, we know that :

$$\forall \alpha \in \mathcal{P}, \ \forall N \in \mathbb{N}, \qquad \sum_{\mathcal{D} \in \mathscr{E}^+(\alpha, N)} \mathbb{P}(\mathcal{D}|\sigma^{\text{ESBAS}}) = 1. \tag{37}$$

By applying recursively Assumption 2, one demonstrates that:

$$\sum_{\mathcal{D} \in \mathscr{E}^+(\alpha, N)} \mathbb{P}(\mathcal{D}|\sigma^{\text{ESBAS}})\mathbb{E}\mu^\alpha_{sub^\alpha(\mathcal{D})} \geq \sum_{\mathcal{D} \in \mathscr{E}^N} \mathbb{P}(\mathcal{D}|\sigma^\alpha)\mathbb{E}\mu^\alpha_\mathcal{D}, \tag{38a}$$

$$\sum_{\mathcal{D} \in \mathscr{E}^+(\alpha, N)} \mathbb{P}(\mathcal{D}|\sigma^{\text{ESBAS}})\mathbb{E}\mu^\alpha_{sub^\alpha(\mathcal{D})} \geq \mathbb{E}_{\sigma^\alpha}\left[\mathbb{E}\mu^\alpha_{\mathcal{D}^{\sigma^\alpha}_N}\right]. \tag{38b}$$

One also notices the following piece-wisely domination from applying recursively Assumption 1:

$$(1-\delta_\tau)\mathbb{E}\mu^*_\infty - \sum_{\mathcal{D} \in \mathscr{E}^{\tau-1}_{3K}} \mathbb{P}(\mathcal{D}|\sigma^{\text{ESBAS}})\mathbb{E}\mu^*_{sub^*(\mathcal{D})} = \sum_{\mathcal{D} \in \mathscr{E}^{\tau-1}_{3K}} \mathbb{P}(\mathcal{D}|\sigma^{\text{ESBAS}}) \left( \mathbb{E}\mu^*_\infty - \mathbb{E}\mu^*_{sub^*(\mathcal{D})} \right), \tag{39a}$$

$$\begin{aligned}
&(1-\delta_\tau)\mathbb{E}\mu^*_\infty \\
&- \sum_{\mathcal{D} \in \mathscr{E}^{\tau-1}_{3K}} \mathbb{P}(\mathcal{D}|\sigma^{\text{ESBAS}})\mathbb{E}\mu^*_{sub^*(\mathcal{D})} \leq \sum_{\mathcal{D} \in \mathscr{E}^+(\alpha^*, \lfloor \frac{\tau-1}{3K} \rfloor) \& |\mathcal{D}| \leq \tau-1} \mathbb{P}(\mathcal{D}|\sigma^{\text{ESBAS}}) \left( \mathbb{E}\mu^*_\infty - \mathbb{E}\mu^*_{sub^*(\mathcal{D})} \right),
\end{aligned} \tag{39b}$$

$$\begin{aligned}
&(1-\delta_\tau)\mathbb{E}\mu^*_\infty \\
&- \sum_{\mathcal{D} \in \mathscr{E}^{\tau-1}_{3K}} \mathbb{P}(\mathcal{D}|\sigma^{\text{ESBAS}})\mathbb{E}\mu^*_{sub^*(\mathcal{D})} \leq \sum_{\mathcal{D} \in \mathscr{E}^+(\alpha^*, \lfloor \frac{\tau-1}{3K} \rfloor)} \mathbb{P}(\mathcal{D}|\sigma^{\text{ESBAS}}) \left( \mathbb{E}\mu^*_\infty - \mathbb{E}\mu^*_{sub^*(\mathcal{D})} \right),
\end{aligned} \tag{39c}$$

$$(1 - \delta_\tau)\mathbb{E}\mu_\infty^* - \sum_{\mathcal{D} \in \mathscr{E}_{3K}^{\tau-1}} \mathbb{P}(\mathcal{D}|\sigma^{\text{ESBAS}})\mathbb{E}\mu_{sub^*(\mathcal{D})}^* \le \mathbb{E}\mu_\infty^* - \sum_{\mathcal{D} \in \mathscr{E}^+(\alpha^*, \lfloor \frac{\tau-1}{3K} \rfloor)} \mathbb{P}(\mathcal{D}|\sigma^{\text{ESBAS}})\mathbb{E}\mu_{sub^*(\mathcal{D})}^* \cdot$$

(39d)

Then, by applying results from Equations 38b and 39d into Equation 36b, one obtains:

$$\mathbb{E}\mu_\infty^* - \mathbb{E}_{\sigma^{\text{ESBAS}}}\left[\mathbb{E}\mu_{sub^*(\mathcal{D}_{\tau-1})}^*\right] \le \mathbb{E}\mu_\infty^* - \mathbb{E}_{\sigma^*}\left[\mathbb{E}\mu_{\mathcal{D}_{\lfloor \frac{\tau-1}{3K} \rfloor}^{\sigma^*}}^\alpha\right] + \delta_\tau\left(\mathbb{E}\mu_\infty^* - \frac{R_{min}}{1-\gamma}\right). \quad (40)$$

Next, the terms in the first line of Equation 32 are bounded as follows:

$$T\mathbb{E}\mu_\infty^* - \mathbb{E}_{\sigma^{\text{ESBAS}}}\left[\sum_{\tau=1}^T \mathbb{E}\mu_{sub^*(\mathcal{D}_{\tau-1})}^*\right] \le T\mathbb{E}\mu_\infty^* - \mathbb{E}_{\sigma^*}\left[\sum_{\tau=1}^T \mathbb{E}\mu_{\mathcal{D}_{\lfloor \frac{\tau-1}{3K} \rfloor}^{\sigma^*}}^\alpha\right] \quad (41a)$$
$$+ \sum_{\tau=1}^T \delta_\tau\left(\mathbb{E}\mu_\infty^* - \frac{R_{min}}{1-\gamma}\right),$$

$$T\mathbb{E}\mu_\infty^* - \mathbb{E}_{\sigma^{\text{ESBAS}}}\left[\sum_{\tau=1}^T \mathbb{E}\mu_{sub^*(\mathcal{D}_{\tau-1})}^*\right] \le \frac{T}{\lfloor \frac{T}{3K} \rfloor}\overline{\rho}_{abs}^{\sigma^*}\left(\lfloor \frac{T}{3K} \rfloor\right) + \left(\mathbb{E}\mu_\infty^* - \frac{R_{min}}{1-\gamma}\right)\sum_{\tau=1}^T \delta_\tau. \quad (41b)$$

Again, for $T \ge 9K^2$:

$$T\mathbb{E}\mu_\infty^* - \mathbb{E}_{\sigma^{\text{ESBAS}}}\left[\sum_{\tau=1}^T \mathbb{E}\mu_{sub^*(\mathcal{D}_{\tau-1})}^*\right] \le (3K+1)\overline{\rho}_{abs}^{\sigma^*}\left(\frac{T}{3K}\right) + \left(\mathbb{E}\mu_\infty^* - \frac{R_{min}}{1-\gamma}\right)\sum_{\tau=1}^T \delta_\tau. \quad (42)$$

Regarding the first term in the second line of Equation 32, from applying recursively Assumption 2:

$$\mathbb{E}_{\sigma^{\text{ESBAS}}}\left[\mathbb{E}\mu_{sub^*(\mathcal{D}_\tau)}^*\right] \le \mathbb{E}_{\sigma^{\text{ESBAS}}}\left[\mathbb{E}\mu_{\mathcal{D}_\tau}^*\right], \quad (43a)$$

$$\mathbb{E}_{\sigma^{\text{ESBAS}}}\left[\mathbb{E}\mu_{sub^*(\mathcal{D}_\tau)}^*\right] \le \mathbb{E}_{\sigma^{\text{ESBAS}}}\left[\max_{\alpha \in \mathcal{P}} \mathbb{E}\mu_{\mathcal{D}_\tau}^\alpha\right]. \quad (43b)$$

From this observation, one directly concludes the following inequality:

$$\mathbb{E}_{\sigma^{\text{ESBAS}}}\left[\sum_{\tau=1}^T \mathbb{E}\mu_{sub^*(\mathcal{D}_\tau)}^*\right] - \mathbb{E}_{\sigma^{\text{ESBAS}}}\left[\sum_{\tau=1}^T \mathbb{E}\mu_{\mathcal{D}_\tau}^{k_\tau}\right] \le \mathbb{E}_{\sigma^{\text{ESBAS}}}\left[\sum_{\tau=1}^T \max_{\alpha \in \mathcal{P}} \mathbb{E}\mu_{\mathcal{D}_\tau}^\alpha - \sum_{\tau=1}^T \mathbb{E}\mu_{\mathcal{D}_\tau}^{k_\tau}\right], \quad (44a)$$

$$\mathbb{E}_{\sigma^{\text{ESBAS}}}\left[\sum_{\tau=1}^T \mathbb{E}\mu_{sub^*(\mathcal{D}_\tau)}^*\right] - \mathbb{E}_{\sigma^{\text{ESBAS}}}\left[\sum_{\tau=1}^T \mathbb{E}\mu_{\mathcal{D}_\tau}^{k_\tau}\right] \le \overline{\rho}_{ss}^{\sigma^{\text{ESBAS}}}(T). \quad (44b)$$

Injecting results from Equations 42 and 44b into Equation 32 provides the result:

$$\overline{\rho}_{abs}^{\sigma^{\text{ESBAS}}}(T) \le (3K+1)\overline{\rho}_{abs}^{\sigma^*}\left(\frac{T}{3K}\right) + \overline{\rho}_{ss}^{\sigma^{\text{ESBAS}}}(T) + \left(\mathbb{E}\mu_\infty^* - \frac{R_{min}}{1-\gamma}\right)\sum_{\tau=1}^T \delta_\tau. \quad (45)$$

We recall here that the stochastic bandit algorithm $\Xi$ was assumed to guarantee to try the best algorithm $\alpha^*$ at least $T/K$ times with high probability $1 - \delta_T$ and $\delta_T \in \mathcal{O}(T^{-1})$. Now, we show that at any time, the longest stochastic bandit run (*i.e.* the epoch that experienced the biggest number of pulls) lasts at least $N = \frac{\tau}{3}$: at epoch $\beta_\tau$, the meta-time spent on epochs before $\beta_\tau - 2$ is equal to

$\sum_{\beta=0}^{\beta_\tau-2} 2^\beta = 2^{\beta_\tau-1}$; the meta-time spent on epoch $\beta_\tau - 1$ is equal to $2^{\beta_\tau-1}$; the meta-time spent on epoch $\beta$ is either below $2^{\beta_\tau-1}$, in which case, the meta-time spent on epoch $\beta_\tau - 1$ is higher than $\frac{\tau}{3}$, or the meta-time spent on epoch $\beta$ is over $2^{\beta_\tau-1}$ and therefore higher than $\frac{\tau}{3}$. Thus, ESBAS is guaranteed to try the best algorithm $\alpha^*$ at least $\tau/3K$ times with high probability $1 - \delta_\tau$ and $\delta_\tau \in \mathcal{O}(\tau^{-1})$. As a result:

$$\exists \kappa_3 > 0, \quad \overline{\rho}_{abs}^{\sigma^{\text{ESBAS}}}(T) \leq (3K+1)\overline{\rho}_{abs}^{\sigma^*}\left(\frac{T}{3K}\right) + \overline{\rho}_{ss}^{\sigma^{\text{ESBAS}}}(T) + \left(\mathbb{E}\mu_\infty^* - \frac{R_{min}}{1-\gamma}\right) \sum_{\tau=1}^{T} \frac{\kappa_3}{\tau}, \quad (46)$$

$$\exists \kappa > 0, \quad \overline{\rho}_{abs}^{\sigma^{\text{ESBAS}}}(T) \leq (3K+1)\overline{\rho}_{abs}^{\sigma^*}\left(\frac{T}{3K}\right) + \overline{\rho}_{ss}^{\sigma^{\text{ESBAS}}}(T) + \kappa \log(T), \quad (47)$$

with $\kappa = \kappa_3 \left(\mathbb{E}\mu_\infty^* - \frac{R_{min}}{1-\gamma}\right)$, which proves the theorem. $\square$

