# OpenReview forum: "Reinforcement Learning Algorithm Selection"
_ICLR.cc/2018/Conference — Accept (Poster)_

### Official Review · AnonReviewer2 · 2017-11-27
**Right performance measure?**

**Rating:** 6
**Confidence:** 5

**Review:**

SUMMARY
The paper considers a meta-algorithm, in the form of a UCB algorithms, that selects base-learners in a pool of reinforcement learning agents.

HIGH LEVEL COMMENTS
In this paper, T refers to the total number of meta-decisions. This is very different from the total number of interactions with the system that corresponds to D_T=sum_{\tau=1}^T |\epsilon_tau|. Wouldn't it make more sense to optimize regret accumulated on this global time?
The proposed strategy thus seems a bit naive since different algorithms from the set \cal P may generate trajectories of different length.
For instance, one algorithm may obtain relatively high rewards very fast with short trajectories and another one may get slightly higher cumulative rewards but on much longer trajectories.
In this case, the meta-algorithm will promote the second algorithm, while repeatedly selecting the first one would yield higher cumulative reward in total over all decision (and not meta-decision) time steps.
This also means that playing T1 meta-decision steps, where T1>>T, may corresponds to a total number of decision steps sum_{\tau=1}^{T_1} |\epsilon'_tau| still not larger than D_T (where \epsilon'_\tau are other trajectories).
Now the performance of a specific learner with T1 trials may be much higher than with T trials, and thus even though the regret of the meta-learner is higher, the overall performance of the recommended policy learned at that point may be better than the one output with T meta-decisions.

Thus, it seems to me that a discussion about the total number of decision steps (versus meta-deciion steps) is missing in order to better motivate the choise of performance measure, and generates possibly complicated situations, with a non trivial trade-off that needs to be adressed. This also suggests the proposed algorithm may be quite sub-optimal in terms of total number of decision steps.
My feeling is that the reason you do not observe a too bad behavior in practice may be due to the discount factor.

OTHER COMMENTS:
Page 4: What value of \xi do you use in ESBAS ? I guess it should depend on R/(1-\gamma)?

Page 5: "one should notice that the first two bounds are obtained by summming up the gaps": which bounds? which gaps? Can you be more precise?
Next sentence also needs to be clarified. What is the budget issue involved here?

Can you comment on the main reason why you indeed get O(sqrt{T}) and not O(\sqrt{T poly-log(T)}) for instance?

Theorem 3: "with high probability delta_T in O(1/T)": do you mean with probability higher than 1-delta_T, with delta_T = O(1/T) ?

Paragraph on Page 15 : Do you have a proof for the claim that such algorithms indeed satisfy these assumptions ?
Especially proving that assumption 3 holds may not be obvious since one may consider an algorithm may better learn using data collected from its played polocy rather than from other policies.

(14b-c, 15d): should u be u^\alpha ? I may simply be confused with the notations.

DECISION
Even though there seems to be an important missing discussion regarding optimization of performance with respect to the total number of decision steps rather than the total number of meta-decision steps,
I would tend to accept the paper. Indeed, if we left apart the choice for this performance measure, the paper is relatively well written and provides both theoretical and practical results that are of interest. But this has to be clarified.

---

### Official Review · AnonReviewer3 · 2017-11-28

**Rating:** 6
**Confidence:** 3

**Review:**

The authors consider the problem of dynamically choosing between several reinforcement learning algorithms for solving a reinforcement learning with discounted rewards and episodic tasks. The authors propose the following solution to the problem:
- During epochs of exponentially increasing size (this technique is well known in the bandit litterature and is called a "doubling trick"), the various reinforcement learning algorithms are "frozen" (i.e. they do not adapt their policy) and the K available algorithms are sampled using the UCB1 algorithm  in order to discover the one which yields the highest mean reward.

Overall the paper is well written, and presents some interesting novel ideas on aggregating reinforcement learning algorithms. Below are some remarks:

- An alternative and perhaps simpler formalization of the problem would be learning with expert advice (using algorithms such as "follow the perturbed leader"), where each of the available reinforcement learning algorithms acts as an expert. What is more, these algorithms usually yield O(sqrt(T)log(T)), which is the regret obtained by the authors in the worse case (where all the learning algorithms do converge to the optimal policy at the optimal speed O(1/sqrt(T)). It would have been good to see how those approaches perform against the proposed algorithms.
- The authors use UCB1, but they did not try KL-UCB, which is stricly better (in fact it is optimal for bounded rewards). In particular the numerical performance of the latter is usually vastly better than the former, especially when rewards have a small variance.
- The performance measure used by the authors is rather misleading ("short sighted regret"): they compare what they obtain to what the policy discovered by the best reainforcement learning algorithm \underline{based on the trajectories they have seen}, and the trajectories themselves are generated by the choices made by the algorthms at previous time. Ie in general, there might be cases in which one does not explore enough with this approach (i.e one does not try all state-action pairs enough), so that while this performance measure is low, the actual regret is very high and the algorithm does not learn the optimal policy at all (while this could be done by simply exploring at random log(T) times ...).

---

### Official Review · AnonReviewer1 · 2017-11-28
**Fairness assumption unrealistic but unavoidable?**

**Rating:** 7
**Confidence:** 4

**Review:**

The paper considers the problem of online selection of RL algorithms. An algorithm selection (AS) strategy called ESBAS is proposed. It works in a sequence of epochs of doubling length, in the following way: the algorithm selection is based on a UCB strategy, and the parameters of the algorithms are not updated within each epoch (in order that the returns obtained by following an algorithm be iid). This selection allows ESBAS to select a sequence of algorithms within an epoch to generate a return almost as high as the return of the best algorithm, if no learning were made. This weak notion of regret is captured by the short-sighted pseudo regret.

Now a bound on the global regret is much harder to obtain because there is no way of comparing, without additional assumption, the performance of a sequence of algorithms to the best algorithm had this one been used to generate all trajectories from the beginning. Here it is assumed that all algorithms learn off-policy. However this is not sufficient, since learning off-policy does not mean that an algorithm is indifferent to the behavior policy that has generated the data. Indeed even for the most basic off-policy algorithms, such as Q-learning, the way data have been collected is extremely important, and collecting transitions using that algorithm (such as epsilon-greedy) is certainly better than following an arbitrary policy (such as uniformly randomly, or following an even poorer policy which would not explore at all). However the authors seem to state an equivalence between learning off-policy and fairness of learning (defined in Assumption 3). For example in their conclusion they mention “Fairness of algorithm evaluation is granted by the fact that the RL algorithms learn off-policy”. This is not correct. I believe the main assumption made in this paper is the Assumption 3 (and not that the algorithms learn off-policy) and this should be dissociated from the off-policy learning.

This fairness assumption is very a strong assumption that does not seem to be satisfied by any algorithm that I can think of. Indeed, consider D being the data generated by following algorithm alpha, and let D’ be the data generated by following algorithm alpha’. Then it makes sense that the performance of algorithm alpha is better when trained on D rather than D’, and alpha’ is better when trained on D’ than on D. This contradicts the fairness assumption.

This being said, I believe the merit of this paper is to make explicit the actual assumptions required to be able to derive a bound on the global regret. So although the fairness assumption is unrealistic, it has the benefit of existing…

So in the end I liked the paper because the authors tried to address this difficult problem of algorithmic selection for RL the best way they could. Maybe future work will do better but at least this a first step in an interesting direction.

Now, I would have liked a comparison with other algorithms for algorithm selection, like:
- explore-then-exploit, where a fraction of T is used to try each algorithm uniformly, then the best one is selected and played for the rest of the rounds.
- Algorithms that have been designed for curriculum learning, such as some described in [Graves et al., Automated Curriculum Learning for Neural Networks, 2017], where a proxy for learning progress is used to estimate how much an algorithm can learn from data.

Other comments:
- Table 1 is really incomprehensible. Even after reading the Appendix B, I had a hard time understanding these results.
- I would suggest adding the fairness assumption in the main text and discussing it, as I believe this is crucial component for the understanding of how the global regret can be controlled.
- you may want to include references on restless bandits in Section 2.4, as this is very related to AS of RL algorithms (arms follow Markov processes).
- The reference [Best arm identification in multi-armed bandits] is missing an author.

---

### Decision · Program_Chairs · 2018-01-29
**ICLR 2018 Conference Acceptance Decision**

**Decision:**

Accept (Poster)

**Comment:**

The reviewers are unanimous in accepting the paper.  They generally view it as introducing an original approach to online RL using bandit-style selection from a fixed portfolio of off-policy algorithms.  Furthermore, rigorous theoretical analysis shows that the algorithm achieves near-optimal performance.

The only real knock on the paper is that they use a weak notion of regret i.e. short-sighted pseudo regret.  This is considered inevitable, given the setting.